



# Integrative and comprehensive Understanding on Polar Environments (iCUPE): the concept and initial results

Tuukka Petäjä[1], Ella-Maria Duplissy[1], Ksenia Tabakova[1], Julia Schmale[2,3], Barbara Altstädter[4], Gerard Ancellet[5], Mikhail Arshinov[6], Yrii Balin[6], Urs Baltensperger[2], Jens Bange[7], Alison Beamish[8], Boris Belan[6], Antoine Berchet[9], Rossana Bossi[10], Warren R.L. Cairns[11], Ralf Ebinghaus[12], Imad El Haddad[2], Beatriz Ferreira-Araujo[13], Anna Franck[1], Lin Huang[14], Antti Hyvärinen[15], Angelika Humbert[16,17], Athina-Cerise Kalogridis[18], Pavel Konstantinov[19], Astrid Lampert[4], Matthew MacLeod[20], Olivier Magand[21], Alexander Mahura[1], Louis Marelle[5,21], Vladimir Masloboev[22], Dmitri Moisseev[1], Vaios Moschos[2], Niklas Neckel[16], Tatsuo Onishi[5], Stefan Osterwalder[21], Aino Ovaska[1], Pauli Paasonen[1], Mikhail Panchenko[6], Fidel Pankratov[22], Jakob B. Pernov[10], Andreas Platis[7], Olga Popovicheva[23], Jean-Christophe Raut[5], Aurélie Riandet[9], Torsten Sachs[8], Rosamaria Salvatori[24], Roberto Salzano[25], Ludwig Schröder[16], Martin Schön[7], Vladimir Shevchenko[26], Henrik Skov[10], Jeroen E. Sonke[13], Andrea Spolaor[11], Vasileios Stathopoulos[18], Mikko Strahlendorff[15], Jennie L. Thomas[21], Vito Vitale[11], Sterios Vratolis[18], Carlo Barbante[11,27], Sabine Chabrillat[8], Aurélien Dommergue[21], Konstantinos Eleftheriadis[18], Jyri Heilimö[15], Kathy S. Law[5], Andreas Massling[10], Steffen M. Noe[28], Jean-Daniel Paris[9], André Prévôt[2], Ilona Riipinen[20], Birgit Wehner[29], Zhiyong Xie[12] and Hanna K. Lappalainen[1,15]

[1]Institute for Atmospheric and Earth System Research/Physics, Faculty of Science, P.O. Box 64, 00014 University of Helsinki, Finland

[2]Laboratory of Atmospheric Chemistry, Paul Scherrer Institute, Villigen PSI, Switzerland

[3]School of Architecture, Civil and Environmental Engineering, École Polytechnique Fédérale de Lausanne, Switzerland

[4]Institute of Flight Guidance, Technische Universität Braunschweig, Germany

[5]LATMOS/IPSL, Sorbonne Université, UVSQ, CNRS, Paris, France

[6]V.E. Zuev Institute of Atmospheric Optics of Siberian Branch of the Russian Academy of Science (IAO SB RAS), Tomsk, Russia

[7]Center for Applied Geoscience, Eberhard-Karls-University, Tübingen, Germany

[8]GFZ German Research Centre for Geosciences, Telegrafenberg, Potsdam, Germany

[9]Laboratoire des Sciences du Climat et de l'Environnement, CEA-CNRS-UVSQ, IPSL, Gif-sur-Yvette, France

[10]Department of Environmental Science, iClimate, Aarhus University, Frederiksborgvej 399, 4000 Roskilde, Denmark

[11]Institute of Polar Sciences, National Research Council of Italy (CNR), Via Torino 155, (I) 30172 Venezia-Mestre (VE), Italy

[12]Helmholtz-Zentrum Geesthacht (HZG), Centre for Materials and Coastal Research, Geesthacht, Germany

[13]Geosciences Environnement Toulouse, CNRS/IRD/Universite Paul Sabatier, Toulouse, France

[14]Climate Research Division, ASTD/STB, Environment & Climate Change Canada (ECCC), Toronto, Canada

[15]Finnish Meteorological Institute (FMI), Helsinki, Finland

[16]Alfred Wegener Institute Helmholtz Centre for Polar and Marine Research (AWI), Bremerhaven, Germany

[17]Department for Geoscience, University of Bremen, Bremen, Germany

[18]Environmental Radioactivity Laboratory, Institute of Nuclear and Radiological Science & Technology, Energy & Safety, NCSR Demokritos, Athens, Greece



[19]Faculty of Geography, Moscow State University, Moscow

[20]Department of Environmental Science and Analytical Chemistry, Stockholm University, Sweden

[21]Univ. Grenoble Alpes, CNRS, IRD, Grenoble INP, IGE, 38000 Grenoble, France

[22]Institute of Northern Environmental Problems (INEP), Kola Science Center (KSC), Russian Academy of Sciences (RAS), Apatity, Murmansk region, Russia

[23]Skobeltsyn Institute of Nuclear Physics, Lomonosov Moscow State University, Moscow 119991, Russia

[24]Institute of Polar Sciences, National Research Council of Italy (CNR), Monterotondo (RM), Italy

[25]Institute of Atmospheric Pollution Research, National Research Council of Italy (CNR), Sesto Fiorentino (FI), Italy

[26]P.P. Shirshov Institute of Oceanology, Moscow

[27]Ca' Foscari University of Venice, Department of Environmental Sciences, Informatics and Statistics, Via Torino 155, 30172 Mestre, Venice, Italy

[28]Institute of Agricultural and Environmental Sciences, Estonian University of Life Sciences, Tartu, Estonia

[29]Leibniz Institute for Tropospheric Research, Leipzig, Germany



**Abstract.** The role of polar regions increases in terms of megatrends such as globalization, new transport routes, demography and use of natural resources consequent effects of regional and transported pollutant concentrations. We set up the ERA-PLANET Strand 4 project "iCUPE - integrative and Comprehensive Understanding on Polar Environments" to provide novel

insights and observational data on global grand challenges with an Arctic focus. We utilize an integrated approach combining in situ observations, satellite remote sensing Earth Observations (EO) and multi-scale modeling to synthesize data from comprehensive long-term measurements, intensive campaigns and satellites to deliver data products, metrics and indicators to the stakeholders concerning the environmental status, availability and extraction of natural resources in the polar areas. The iCUPE work consists of thematic state-of-the-art research and provision of novel data in atmospheric pollution, local sources

and transboundary transport, characterization of arctic surfaces and their changes, assessment of concentrations and impacts of heavy metals and persistent organic pollutants and their cycling, quantification of emissions from natural resource extraction and validation and optimization of satellite Earth Observation (EO) data streams. In this paper we introduce the iCUPE project and summarize initial results arising out of integration of comprehensive in situ observations, satellite remote sensing and multiscale modeling in the Arctic context.


## 1 Introduction

The project "iCUPE - integrative and Comprehensive Understanding on Polar Environments" is motivated by the increasing role of Arctic regions in terms of megatrends such as globalization, new transport routes, demography and use of natural resources. These megatrends will rapidly and drastically affect on the environment. In particular, the Arctic will face such

grand challenges as the soil and water pollution, climate change, land use change, higher demand for resource extraction, increased anthropogenic emissions due to year-round shipping in the Arctic Ocean and other local sources as well as long-range transported pollution from Europe, Asia and North America (e.g. Buixade Farré et al., 2014). Overall, land and ocean areas located in the polar latitudes are currently undergoing and will undergo substantial changes due to increased anthropogenic activities and shipping during the next decades. These activities will put the fragile Arctic environment and the

population living in this area in a vulnerable position. The changes will pose unpredictable consequences on food chains, biodiversity and the primary production of different plant ecosystems and ecosystem capacity to recover from the pollution exposure and environmental changes (e.g. Arnold et al., 2016).

The future warming of the Arctic will affect demographic trends by increasing urbanization and migration to northern regions, and by accelerating changes in societal issues and air quality (Schmale et al., 2018). One major consequence of warming in

the northern latitudes is related to changes in the cryosphere, including ice sheet mass loss (Helm et al., 2014), the thawing of permafrost and the Arctic Ocean becoming sea ice free part of the year (Kokelj et al., 2017; Meier et al., 2014; Kulmala et al., 2015; Lappalainen et al., 2016; Boy et al., 2019). Even with limiting global warming to 1.5° or 2° C, temperatures over the high Arctic, in particular north Greenland, will rise by 3-4° C by 2100 due to the polar amplification, enhancing impacts like mass loss of the ice sheet (Rückamp et al., 2018). This will accelerate global trade activities in the Arctic region then likely

the northern sea route is seasonally opened for shipping between the Atlantic and Asia's Far East. Northern ecosystems and Arctic regions are a source of major natural resources such as oil, natural gas and minerals. The availability and exploitation of natural resources depends also on how significantly the permafrost thaw will damage existing infrastructure.

Human activities have had a profound impact on the composition of the atmosphere and the pollution in the environment through the introduction of increasing quantities of heavy metals and other trace elements (Barbante et al., 2001), radioactive

nuclides (Ezerinkis et al., 2014), synthesized organic compounds (Hermanson et al., 2010), aerosols such as black carbon (McConnell and Edwards, 2008), trace gases and greenhouse gases. Anthropogenic contaminants can be transported over long distances and accumulate into polar areas. Persistent Organic Pollutants (POPs), such as Polybrominated Diphenyl Ethers



(PBDE), Polycyclic Aromatic Hydrocarbons (PAH), Polychlorinated Biphenyl (PCB) and persistent Contaminants of Emerging Concern (CEC) (Sauve and Desrosiers, 2014) are rarely produced in the Arctic, but have been found in Arctic wildlife, lake sediments, annual snow and ice (Herbert et al., 2005; Ma et al., 2011; Seki et al., 2015). Mercury and other heavy metals, such as As, Cd, and Pb, are considered toxic at any level. Their presence is generally determined by local geochemistry, but they can be emitted by human activities resulting in their increased abundance in the polar areas (Barbante et al., 2001; Zheng et al., 2015; Angot et al., 2016). Black carbon (BC), a fine component of almost pure carbon from incomplete combustion, is able to modify the snow albedo by absorbing incoming solar radiation (Bond et al., 2013; Jiao et al., 2014). Human activities are impacting the net abundance of these pollutants in the atmosphere, but there is a lack of data exploring the deposition patterns and the abundance of anthropogenic contaminants in polar areas. We also need an improved understanding about their redistribution into different environmental spheres including the biota of the Arctic and Antarctic and the full life cycle of these pollutants (Wöhrnschimmel et al., 2013).

Local emissions make currently only a small contribution to the atmospheric loadings of various pollutants in polar areas, but this might change in the near future as Arctic ice-free areas will extend and more extensive Arctic shipping will become possible (Corbett et al., 2010; Yumashev et al., 2018). Nevertheless, air pollutants from other areas in the world do reach high Arctic regions and have been estimated to have significant impacts on the regional ecosystem and climate (Di Pierro et al., 2011; Breider et al., 2014). Knowledge about source contribution of atmospheric pollution is very limited and further efforts in terms of detailed source identification are urgently needed to formulate and settle mitigation strategies (e.g. Law et al., 2014). The measurements of short-lived climate forcers (SLCF), and their precursors, are necessary for evaluating the impacts of increased regional and international activities, e.g. in relation to natural resource extraction, especially in fragile Arctic environments. Correspondingly, similar activities need to be carried out in Antarctica, which has a minimal amount of anthropogenic influences and can provide clean reference observations.

The existing observational networks with comprehensive in situ observational capacity for the measurements of atmospheric concentrations of air pollutants extend to Arctic and Antarctic environments (e.g. Uttal et al., 2016). However, there are still large gaps in the current measurement networks (Lappalainen et al., 2016), and the interaction between the networks, made up of different national activities, needs to be improved. Polar activities are often based on national activities and missing synergistic benefits of cooperation in the challenging environments. In summary, the methodology of data acquisition, data quality control and future strategies on data flows and data streams are not harmonized on either the European or global scale. Furthermore, particularly in the polar areas, measurements are not always continuous but often carried out campaign-wise due to economic, environmental or logistical challenges.

Satellite remote sensing in the Arctic is based on active and passive missions of varying spatial resolution, repeat visit times and coverage of high latitudes. Monitoring the surface properties and its variations in the Arctic region is a powerful tool to assess the impacts of changes induced into this vulnerable environment. The distribution of different types of land cover (snow, ice, vegetation, soil) can be efficiently analyzed using optical data obtained from the new satellite missions merged with data collected during field campaigns and data acquired from cooperative observer networks. In particular, different patterns of snow cover (as well as soil and vegetation cover) exercise considerable influence on the surface energy balance, since variations in land cover change the surface albedo. While there are established methods for retrieving basic variables, changing snow and ice surfaces and ice-free areas are still challenging and the large variability of system itself is limiting the accuracy of such retrievals (Bokhorst et al., 2016). New Sentinel series of Copernicus sensors in the orbit make it possible to retrieve improved land surface variables due to increased capacities in terms of spatial, temporal, spectral and angular observations. As a consequence, new Earth Observation (EO) technics will lead to multi-mission time series needed for data assimilation into models of the Earth system compartments in the Arctic.

In summary, in order to address the current state of the environment in the polar areas and to provide fact-based decision-making tools for the society in the future, comprehensive high-quality observations of atmospheric concentrations of aerosols,



trace gases and related environmental variables from in situ observations are required in concert with EO from space (Petäjä et al., 2014; Hari et al., 2016). The EO data can be used to study the interactions between different types of surfaces and the atmosphere. The results obtained allow us to evaluate the impact of pollutants on the equilibrium of the Arctic system and provide an important input for the evolutionary scenarios of Arctic environment. The picture needs to be harmonized and

supported with complementary multi-scale modeling (e.g. Kulmala et al., 2011a; Kulmala, 2018). The need to establish and maintain long-term, coherent and coordinated observations and research activities on environmental quality and natural resources in polar areas drives iCUPE activities. The core idea of iCUPE is the development of novel, integrated, quality-controlled and harmonized in situ observations and satellite data in the polar areas, as well as supply of data products to the end users. The impact of this integrated Arctic Observing System needs to be demonstrated for changing campaign practices

to continuous monitoring activities. The Sustaining Arctic Observing Networks (SAON) initiative under the auspices of the Arctic Council has been developing an assessment framework for this since 2017 and iCUPE has been helping in making the first full value tree analysis for this. It connects observing activities in situ and from satellites through modeling and services to key objectives of societal benefit areas. This will help greatly in forming impact assessments in the future.

The aim of this paper is to introduce an on-going project iCUPE and summarize its initial results. We put a specific emphasis

on black carbon and persistent pollutants in the Arctic context. We explore snow and ice core samples to put the current concentrations in longer perspective. We underline the capacity of the continuous observations to monitor the impact of policies to reduce the emissions. We showcase the potential to address the pollution in the Arctic environment by integrating satellite remote sensing, airborne observations, in situ data and modeling. The modern comprehensive source apportionment can resolve the different sources of atmospheric aerosols and differentiate between sources within and outside the Arctic

environment. We also discuss the iCUPE impact and relevance for the Arctic research and for the stakeholder communities.

## 2 The concept

The motivation behind iCUPE stems from the need to answer the global environmental challenges in a polar context (Fig. 1). The underpinning concept of iCUPE is that transdisciplinary research utilizing the full capacity of comprehensive in situ observations together with state-of-the-art satellite observations is required to make advances in the understanding of

atmospheric and cryospheric processes in the Arctic environment (Fig. 2). Therefore, the work in iCUPE utilizes both expertise in in situ observations and satellite remote sensing in a close connection to modeling frameworks to answer this need (Fig. 3). The work is closely connected to on-going activities such as Integrated Arctic Observation System (INTAROS, Sandven et al., 2018) and Multidisciplinary drifting Observatory for the Study of Arctic Climate (MOSAiC, Shupe et al., 2018) expedition conducted in polar areas through collaboration. These connections enable the iCUPE consortium partners to facilitate

interactions and strengthen coordination between the national and international activities in the polar areas. The wide spectrum of observational quantities, data products and modeled variables are required to enable the delivery of integrated data required for decisions related to the Arctic pollution.

## 3 iCUPE initial results

### 3.1 Atmospheric observation capacity in the Arctic

Figure 3 depicts year-round monitoring aerosol measurements in the Arctic. Within the International Arctic Systems for Observing the Atmosphere, IASOA (Uttal et al., 2016), there are 14 stations all around the Arctic with a clear prevalence of the west longitudes. In addition to the IASOA sites, Stations for Measuring Ecosystem – Atmosphere Relations (SMEAR, Hari et al., 2016) observation network extends to the Arctic as the SMEAR I in Värriö, Finnish Lapland, has provided aerosol and trace gas observations since 1992 (Hari et al., 1994). Many of these observation sites contribute to the World





Meteorological Organization's Global Atmospheric Watch (WMO-GAW) as well as to many thematic European Research infrastructures, such as Integrated Carbon Observation System (ICOS), Aerosols, Clouds and Trace Gases Research Infrastructure (ACTRIS). Regionally Svalbard Integrated Arctic Observing System (SIOS) provides a platform for comprehensive measurement activities within Svalbard area. Overall, the length and breadth of monitoring programs vary greatly from site to site. Some of these sites have been operational for almost 30 years now, measuring a large number of

parameters, while others, such as Cape Baranova, have only been open for the past 2–3 years. A historical perspective of aerosol measurements in the Arctic (and Antarctica) is presented by several review papers via the POLAR-AOD network (Tomasi et al., 2007, 2012, 2015) and Pan Eurasian Experiment (PEEX, Lappalainen et al., 2018; Vihma et al., 2019).

A geographically representative distribution of measurements is of particular importance. The atmospheric in situ observation network in the Arctic has developed during the last three decades by individual scientists and groups utilizing the resources

available. National interests and logistical possibilities have played a role in the initial establishment of the stations and their maintenance. Considering all this, it is somewhat surprising to see that the distribution of measurement sites covers the entire Arctic circle with a relative high homogeneity. Few gaps can be identified in the Canadian coast and in Russian Arctic along the eastern edge of Siberia and Russian Far East (Petäjä et al., 2019).

The coastline of the Russian Arctic is over 24 000 kilometers long, thus being a significant region where more comprehensive

atmospheric and aerosol observations are needed. Within the PEEX program, we have performed a gap analysis in terms of atmospheric and environmental observations within the Russian Arctic (Alekseychik et al., 2016). The work is on-going and the PEEX catalogue consists of metadata from 59 stations altogether at the moment. The stations are operated by universities, Russian Academy of Sciences and Roshydromet. The most comprehensive stations in the Russian Arctic providing atmospheric observations and aerosol data are The Tiksi Hydrometeorological Observatory, Russian Far East (71.6° N, 128.9°

E) and the Cherskii, the Northeast Science Station (68.73° N, 161.38° E).

### 3.2 Results from long-term atmospheric in situ observations

As an example, we present results from long-term observations of atmospheric aerosol particle measurements at the high Arctic site Villum Research Station (VRS) in North Greenland, and long-term data of black carbon data from Mt Zeppelin at Svalbard. Furthermore, we summarize recent mass spectrometric analyses of organic matter collected on filters within the

Arctic during iCUPE and present results from vertical measurements of aerosol particle number concentrations

### 3.2.1 Aerosol particle number concentrations at the high Arctic site Villum Research Station (VRS) in North Greenland

Particle number size distributions (PNSDs) have been continuously measured at VRS from July 2010 onwards. The aerosol number size distributions were determined with a Scanning Mobility Particle Sizer (SMPS) in the size range from 9 to 915 nm following standardization presented in Wiedensohler et al. (2012).

The long-term data shows that the seasonality of the atmospheric aerosol number concentration at VRS is largely governed by synoptic weather patterns. Particularly important is the location of the Arctic Front, which describes the boundary between the cold Arctic air masses and the mid-latitude atmosphere. The Arctic Front effectively blocks atmospheric transport from the mid-latitudes to the high Arctic (e.g. Law and Stohl, 2007). There is considerable variability on the location of the Arctic Front geographically and seasonally (e.g. Klonecki et al., 2003). This leads to a variable contribution between local emissions and

the long-range transported aerosols consequently affecting the observed aerosol number size distribution at VRS (Nguyen et al., 2016). The described seasonality is evident from Fig. 4, which displays the monthly averaged particle number concentration in the accumulation mode (Fig. 4a) and ultrafine modes (Fig. 4b).

Our results from VRS show that in the wintertime (October–April), efficient meridional transport coupled with inefficient wet removal processes permits emissions from anthropogenic sources in the mid-latitudes to reach VRS (Freud et al., 2017). This

Arctic Haze is characterized by high concentrations of sulfate, black carbon, and accumulation mode (particle diameter > 100





nm) aerosol particles (Fig. 4a). The expansion of the Polar Front and inefficient wet removal in the High Arctic allows for the transport and build-up of relatively high concentrations of accumulation mode particles at VRS. This mode increases in concentration from November until it reaches a maximum in April. When the Polar Front retreats, anthropogenic emissions are no longer able to penetrate into the High Arctic. The transition from spring to summer, occurring in May, is accompanied

by a fast decrease in accumulation mode aerosol. The following summer months exhibit a minimum in concentration of this aerosol type (Fig. 4a). This pattern is similarly observed for other anthropogenic pollutants such as sulfate and black carbon (Massling et al., 2015; Lange et al., 2018).

The decrease in Arctic Haze, coupled with increases in sunlight and the melting of sea ice/snowpack, allows natural sources of aerosols and their corresponding precursor gasses to become an important source to the atmospheric burden of Arctic

aerosols. During the summer months, local and regional emissions from natural sources govern the aerosol burden (Barrie, 1986). There is a minimum of ultrafine mode aerosol concentrations in the winter months followed by an increase in April/May, reaching a maximum in July, before decreasing during August/September until reaching a minimum again in October. The aerosol and precursor sources for the ultrafine aerosols (aerosol particle diameter < 100 nm) particularly during the summer can arise from primary marine aerosols as well as secondary sources such as gas-to-particle conversion of precursor

gases (e.g. dimethyl sulfide, oxygenated volatile organic compounds, and halogenated compounds) (Dall 'Osto et al., 2018a; Giamarelou et al., 2016; Mungall et al., 2017; Sipilä et al., 2016).

Due to anthropogenically induced climate change, Arctic sea ice extent is rapidly decreasing as well as thickness with the proportion shifting from multi-year ice to first-year ice (IPCC, 2014). There are also indications for a high contribution of Atlantic warm water contributions to the sea ice loss in the Arctic (Polyakov et al., 2017). This has consequences for the

atmospheric burden of ultrafine mode aerosols or their precursors originating during the summer from open waters, ice edges or open leads. In general, these changes are expected to impact on precursor gas emission rates and therefore new particle formation events (Dall 'Osto et al., 2018), which implies that the clean Arctic lower troposphere may have a large impact on the radiative balance if the newly formed particles reach accumulation mode sizes and therefore act as Cloud Condensation Nuclei (CCN). Elucidating the impact of sea ice loss to the aerosol population are needed to understand the radiative forcing

of Arctic aerosol in the future. Furthermore, climate change may impact on the Arctic atmospheric circulation systems and consequently change the atmospheric transport patterns or intensity of those with respect to the occurrence of accumulation mode aerosol in late winter and early spring in the High Arctic (Arnold et al., 2016). The long-term observation record on the atmospheric aerosol number size distribution in the Arctic will enable us to resolve these interactions and feedbacks in the future.

### 245 3.2.2 Black Carbon concentrations at Mt Zeppelin, Svalbard

Black Carbon (BC) is one of the key short-lived climate forcers contributing to the warming of the Arctic both by absorbing the solar radiation but also by enhancing snow and ice melt by surface deposition (e.g. Bond et al., 2013). As part of the atmospheric observations, the ACTRIS and IASOA networks (Fig. 3) operates a network of aethalometers to determine the atmospheric concentration of BC in the air (Uttal et al., 2016). The quality assurance of data for the equivalent black carbon

concentration (eBC) and the corresponding aerosol absorption coefficient has greatly improved over the last years. Compensation schemes for measurement artifacts as well as harmonization of data obtained by different instruments have been established (Backman et al., 2017) and are continuously updated (Kalogridis et al., 2019).

The results from long-term observations at Zeppelin have been discussed and assessed in several works elaborating on the climatology of BC in the Arctic i.e. (Eleftheriadis et al., 2009.; Sharma et al., 2013; Breider et al., 2017; Schmeisser et al.,

2018). The results presented here is the longest continuous eBC reported record by a single instrument in the High Arctic (Torseth at al., 2019). As an example of long-term observations of BC, we show the latest equivalent BC concentration time series from Zeppelin Station at Svalbard (Fig. 5). The results show a continued gradual reduction in the annual mean value of



observed eBC, while the time series is strongly modulated by a seasonal cycle well known in the Arctic with minima in the summer and maximum in late winter spring. However, trend analysis for aerosol climatology records need to be practiced with
caution in order to remove the effects of the seasonal cycle.

The long-term data series we present here makes it possible to derive some descriptive statistics. The eBC annual mean value has been reduced from 28 to 12 ng m$^{-3}$ with an average reduction of 7 ng m$^{-3}$ per decade. Minimum values over the summer often drop below the detection limits of the instrument while maximum values vary greatly with their occurrence usually related to large scale biomass burning events across Siberia and Alaska. The continuous reduction in fossil fuel usage is a
reason for this reduction but it is well known that emissions are not uniformly changing on a global scale or at least in the Northern hemisphere (Evangeliou et al., 2018)

The long-term observations can provide important insight into how abatement strategies for emission reductions or anthropogenic activities enhanced in certain areas can be responsible for the observed changes. Reverse air mass transport modelling using the FLEXPART model (Stohl et al., 2005) can provide the means to combine eBC observations at a remote
station like Zeppelin and the source areas globally. We simulate the backward transport of a Black Carbon (BC) and an air tracer at the arctic station of Zeppelin during a period representative of the high measured concentrations (spring of 2012). The simulation is driven by reanalysis meteorological inputs from the European Center for Medium-range Weather Forecasts (ECMWF) on a resolution of one degree. The Potential Source Contribution Function (PSCF) is applied on both tracers. Western Siberia appears as the main source region on the PSCF analysis.

Differences in emission sensitivities between the two tracers stress the importance of deposition mechanisms in aerosol transport. Wet scavenging occurs in the presence of clouds and precipitation. Wet deposition is determined from a scavenging coefficient, which is also dependent on the precipitation rate. The sub-grid variability of precipitation rate is calculated as an area fraction in each grid that experiences precipitation. Both large scale and convective precipitation rates are accounted for. We observed that the difference in areal distribution of source areas, the potential significance of the key emission areas to
contribute to eBC in the High Arctic and their estimated climate impact is very much dependent on the microphysical parameterization of the model since the air tracer result is considered a simplified representation of the transport and the metrics above greatly affect the result we observe, when the BC tracer parameterization is applied.

### 3.3 Results from targeted atmospheric field studies in the Arctic

#### 3.3.1 Mass spectrometric measurements for offline source apportionment based on Arctic organic aerosols

Organic compounds are of high importance because they contribute between one and two thirds to the submicron aerosol mass in the Arctic (Willis et al., 2018; Schmale et al., 2018; Popovicheva et al., 2019) and may be co-emitted or interact with other aerosol species, such as black carbon (AMAP, 2015), sulfate (Kirpes et al., 2018) and metals (Shaw et al., 2010). If imported from lower latitudes, they also act as a vehicle of transport for Persistent Organic Pollutants (POPs) to the Arctic (Westgate et al., 2013). Organic aerosols (OA) also absorb (Moschos et al., 2018) and scatter light, thereby changing the radiative balance
(Myhre et al., 2013) and may act as cloud condensation nuclei. OA might become increasingly important in a warming Arctic due to anthropogenic activities (Schmale et al., 2018) and natural emissions, e.g., as a result of expanded vegetation (Bhatt et al., 2010), intensified wildfires (Warneke et al., 2010), decreasing sea ice extent and thickness leading to higher release of marine volatile organic compounds (Mungall et al., 2017), and thawing tundra soils (permafrost) along shores and rivers (Peñuelas et al., 2014; Kramshøj et al., 2018). The continuous monitoring of organic carbon (OC) along with a detailed
chemical analysis to determine its natural and anthropogenic sources, seasonal variability and inter-annual evolution in the Arctic is of prime importance for improved climate simulations and a realistic assessment of the effectiveness of potential mitigation or adaptation actions.

**Offline mass spectrometric aerosol analysis**



The OA chemical composition and corresponding sources remain largely unknown, partly due to the challenging measurement
conditions (Uttal et al., 2016; Kulmala, 2018). For example, tremendous effort is required for the deployment of online aerosol
mass spectrometry at various environments for long time periods. To overcome this challenge an offline Aerodyne aerosol
mass spectrometer (AMS) technique has been introduced based on re-aerosolized liquid filter extracts (Daellenbach et al.,
2016). The method is capable of covering broad spatial and seasonal observations as well as determining the sources of OA
(e.g. primary versus secondary, biogenic versus anthropogenic) (Moschos et al., 2018; Bozzetti et al., 2017). This is achieved
with positive matrix factorization, a bilinear un-mixing receptor model used to describe the input mass spectra time series as
a linear combination of static OA source (factor) profiles and their time-dependent contributions to the total OA loading
(Canonaco et al., 2013). Within iCUPE, we aim to extend the coverage of this technique to the most climate change sensitive
region worldwide. The offline AMS analysis will be combined for the first time with ultra-high-resolution mass spectrometry
coupled with liquid chromatography, for a two-dimensional molecular identification of primary aerosol tracers and secondary
organic aerosol precursors.

**Sampling sites**

We have collected quartz fiber filter samples around the Arctic. This unparalleled effort is expected to produce data for the
chemical and source characterization of OA at nine sites within six countries (Pallas - Finland, Zeppelin and Gruvebadet -
Norway, Villum - Greenland, Alert - Canada, Barrow/Utqiagvik - USA, Tiski and Cape Baranova - Russia) from 68° N to 83°
N (Fig. 3) covering the period from 2014 to 2019. These include both coastal/archipelago high Arctic stations and boreal forest
Eurasian sites near the Arctic Circle with different emission exposure characteristics. Here we show offline AMS data for
samples collected at two stations: Ice Base Cape Baranova (Russia) and Alert (Canada).

The research station Cape Baranova was built in 2013.  It is located near the cape Baranova, on the coast of the Shokal Strait
which divides Bolshevik and October Revolution islands of the Archipelago Severnaya Zemlya (79.16° N, 101.45° E). The
area adjacent to the station is characterized by the presence of sea ice, dome-shaped glaciers and icebergs. Polar night and day
at the station Cape Baranova last from October 22 to February and from April 22 to August 22, respectively. The air
temperature in summer (June–August) is from 0 to 4° C. In winter (October–April) temperatures range from -25 to -45° C. The
area of the station is characterized by stable winds mainly from southern direction with an average speed of 10-15 m s⁻¹. During
the transition periods of the year, the wind speed can reach 50 m s⁻¹.

Alert was opened in 1986 as Canadian's first research station for the continuous monitoring of background concentrations of
trace gases and aerosols. It is located at 82.5° N, 62.37° W (210 m a.s.l.) with prevailing winds from the southwest, which
usually bring clear skies and warmer temperatures. North winds off the ocean are typically accompanied by fog and sudden
drops in temperature.

**Preliminary results from the filter sampling**

In Fig. 7, we show offline AMS-based fragment relative contributions of the different organic families in October 2015
(transition to polar night), for the station Cape Baranova (a) and for Alert (b). Even though only limited conclusions can be
drawn from the analyses of single filter samples, there is considerable variability in the composition of organic fragments
between the two samples. During the same autumn period, the 2-day Cape Baranova filter sample contains more than one third
hydrocarbon-like fragments (family CH) and about 50 % oxygenated species (families CHO, CHOgt1). The 2-week Alert
filter sample contains more, i.e. roughly 90 %, and more strongly oxygenated fragments (especially mass to charge ratio m/z
44, family CHOgt1). This can be an indication for the extreme remoteness of the site, because OA has to be advected over
long distances before it reaches Alert and can hence be oxidized during transport.

The difference spectrum for Cape Baranova (Fig 7c) emphasizes potential seasonal differences in fragments which have
relatively high overall contribution to the organic mass. Specifically, the relative abundance of N-containing fragments is
indicative for May (2-day filter), whereas the October sample contains more oxygenated fragments. The former might
potentially be linked to transported anthropogenic (e.g., fossil fuel) emissions. No marked significant differences are observed



for the CH fragments. Ongoing analyses including the other Arctic stations suggest significant variability among the different sites and seasons in the relative fraction of fragments that are markers of certain sources, indicating largely regionally specific sources of OA across the Arctic land surface.

The data analysis is in its early stages but we anticipate a number of outcomes that are valuable for the upcoming assessment by the Expert Group on Short-Lived Climate Forcers of the Arctic Monitoring and Assessment Programme (AMAP), future ultra-high-resolution mass spectrometric measurements of OA in the Arctic, comparison of atmospheric OA with OA in ice cores for historical trends of sources and composition.

**3.3.2 Vertical measurements of aerosol particle number concentrations at Svalbard**

Ground based measurements are performed continuously at specific sites around the Arctic, but vertical measurements are still rare. To investigate the spatial distribution of aerosol particles in an Arctic environment a four-week measurement campaign was conducted with two different types of unmanned aerial systems (UAS) between April and May in 2018 in Ny-Ålesund (Spitsbergen, Norway). The UAS ALADINA (see Fig. 8a) is equipped with miniaturized aerosol instrumentation (two condensation particle counters with different lower detection limits, optical particle spectrometer, aethalometer) and

meteorological sensors (Altstädter et al., 2015). ALADINA was operated from ground to 800 m a.g.l. with flight times of up to 40 min. A second fixed-wing UAS of type MASC3 (Fig. 8b) was operated in parallel focusing on meteorological measurements. MASC3 probed the lower atmosphere from around 20 to 600 m a.g.l. with flight times of up to 1.5 hours. It carried a sensor payload for measuring turbulent quantities of pressure, temperature, humidity, and the 3D wind vector, providing 100-Hz data (Rautenberg et al., 2019).

ALADINA flights were focussing on aerosol profiles while MASC3 flew at several fixed altitudes and over a large horizontal extent with high spatial resolution (less than one meter for the wind vector and the temperature). The 3D data series show the spatial horizontal and vertical variability of the different layers, which enable characterizing the complex flow properties in the fjord around Ny-Ålesund.

The resulting data also provides a spatial variability of turbulent properties (sensible heat, turbulent kinetic energy, momentum

flux). The UAS data improve the understanding of sources and transport processes of aerosol particles in the Arctic, link the observations close to the fjord and at the Zeppelin mountain station (Ström et al., 2009; Tunved et al., 2013), and help to detect regions where new particle formation takes place. Around 200 vertical profiles were performed with the UAS ALADINA between ground and 850 m a.g.l., thus connecting the ground-based measurements at Gruvebadet laboratory, located near the village of Ny-Ålesund and the data sampled at the Zeppelin observatory at the height of 474 m a.s.l. Further measurement

flights were operated horizontally above snow cover and above open water in order to capture the possible impact of biogenic activity on the NPF, as shown along the flight path in Fig. 9.

Figure 10 shows selected profiles of preliminary data of one selected measurement day. The potential temperature and water vapor mixing ratio profiles show that the boundary layer at Svalbard consists of a shallow unstable layer with elevated humidity at the surface. This layer grows higher during the day. However, even at 11:17 UTC the mixing reaches only 150 m and higher

than that, the boundary layer remains stably stratified. Number concentration of accumulation mode particles remained rather constant as a function of height. However, nanoparticles between the sizes of 4–12 nm varied and several distinct layers were identified. The nanoparticle concentrations were the highest at the elevation closest to the surface, but high number concentrations were observed at altitudes up to 500 m. High nanoparticle concentrations in the residual layer has been observed in different environments as well (e.g. Wehner et al., 2010; Altstädter et al., 2018, Leino et al., 2019; Carnerero et al., 2019).

Further analysis needs to be done to conclude if this is caused by anthropogenic pollution or locally restricted NPF events. If NPF events occur on such small scales vertical measurements of aerosol particles are needed to estimate the contribution of such small particles on the regional aerosol balance because many of these layers are not captured by the ground-based measurement sites.





### 3.4 Mercury in the Arctic

High concentrations of mercury (Hg) in Arctic biota pose a threat to local populations and wildlife (Douglas et al., 2012; AMAP, 2012). The scarce anthropogenic Hg emission sources in the Arctic have left scientists to wonder how mid-latitude emissions reach the Arctic Ocean marine ecosystem (e.g. Durnford et al., 2010). The discovery of massive Arctic atmospheric Hg depletion events in 1998 (Schroeder et al., 1998) that are associated with sea-ice derived reactive halogen oxidants (Skov et al., 2004), have fueled a paradigm where mid-latitude urban-industrial Hg emissions reach the Arctic exclusively via the

atmosphere. However, subsequent research has shown that 70-80 % of the deposited Hg is photochemically reemitted back to the atmosphere only hours after deposition (Obrist et al., 2017; Brooks et al., 2006). In iCUPE, we performed observations on the Hg concentrations in the Arctic cryosphere, explored the role of river systems to the Arctic Ocean mercury load and explored the interactions between halogen compounds and Hg.

### 3.4.1 Arctic Mercury cycle

In 2012, a coupled 3-D Ocean-atmosphere model of the arctic Hg cycle suggested that a source of Hg to the Arctic Ocean was missing (Fisher et al., 2012). The missing source was suggested to be Arctic Rivers, in particular Russian rivers that account for 80 % of river run-off to the Arctic Ocean. We monitored year-round Hg levels in the Yenisei and Severnaya Dvina rivers from 2012 to 2016 and confirm that Russian rivers transport large amounts of Hg to the Arctic Ocean (Sonke et al. 2018, Fig. 11b). As part of the iCUPE project these results were integrated into a 3D Arctic mercury model, developed by co-workers

from Harvard University (Sonke et al., 2018). We showed that anthropogenic Hg emissions from mid-latitude industrial sources do not directly reach the Arctic Ocean ecosystem. Instead, atmospheric elemental $Hg^0$ is taken-up year-round by arctic tundra vegetation and soils (Obrist et al., 2017; Jiskra et al., 2018). Springtime snow melt mobilizes the tundra soil mercury, which is bound to plant-derived carbon, via rivers to the Arctic Ocean, where it becomes partly available to the marine food web.

As a summary, the 3D model suggests that a large portion of riverine Hg is photochemically reduced in the surface Arctic Ocean and emitted into the atmosphere. The river Hg budget, together with recent observations on tundra Hg uptake and Arctic Ocean Hg dynamics, provide a consistent view of the Arctic Hg cycle where continental ecosystems traffic anthropogenic Hg emissions to the Arctic Ocean via rivers, and where the Arctic Ocean exports Hg to the atmosphere, to the Atlantic Ocean, and to marine sediments (Sonke et al., 2018, Fig. 11). Recent iCUPE research suggests that Arctic warming and permafrost thaw

risk doubling tundra soil Hg run-off to the Arctic Ocean via rivers, thereby potentially increasing health risks to humans (Lim et al., 2019).

### 3.4.2 Mercury deposition to the Svalbard snowpack

To expand the knowledge on the role of snowpack in the Hg life cycle, we performed targeted field studies (Spolaor et al., 2018, 2019) to determine the seasonality of Hg deposition as well the total Hg deposition from the atmosphere to snow

preserved in the Arctic environment in the Svalbard archipelago, specifically in the Spitsbergen region (Fig. 12). The annual snowpack is defined as the snow that accumulates over a glacier surface during the winter. The snow season in Svalbard changes year by year but typically accumulation of snow starts at the end of September and ends at the end of May in concomitance with the temperature rise and the start of snow melting (Spolaor et al., 2016). The snow season is dependent on altitude, and sites at a higher elevation can preserve part of the annual snowpack throughout the year. The equilibrium line,

which is the altitude above which snow accumulated during the snow season is partially preserved and is located at approximately 600 m a.s.l in the Svalbard region, below this height the snow is completely removed.

In this study we selected five locations to study the Hg concentration in the snow pack (Fig. 13). The lower snow pit was dug in the Midtre Lovenbreen glacier (MLB) at an altitude of 401 m a.s.l., the Austre brogerbreen (BRG) snow pit at 484 m a.s.l.,





the snow pit on the EdithBreen (EDB) at 620 m a.s.l., the Kongsvegen (KNG) snow-pit at 710 m a.s.l. and the Holthedalfonna
(HDF) snow-pit at 1100 m a.s.l. The sampling was done following a constant sampling step of 5 cm starting from the top to
the bottom of the snowpack (identified as the glacier icy surface or by the snow preserved from the previous year, the latter
case only for the KNG and HDF sites). The results show that the Hg concentrations (Fig. 13, lower panel) ranged from 0.5 pg
g$^{-1}$ up to 5 pg g$^{-1}$ with an average concentration of 1.5 pg g$^{-1}$. The higher concentrations were detected at the EDB location
(2.00 ± 0.56 pg g$^{-1}$). The MLB and BRG had a concentration of 1.69 ± 1.04 pg g$^{-1}$ and 1.72 ± 0.76 pg g$^{-1}$, respectively. The
lower mean concentration was found in the HDF and KNG snowpack with a concentration of 0.80 ± 0.69 pg g$^{-1}$ and 1.38 ±
0.61 pg g$^{-1}$, respectively. The snowpack sampled approximately at or below 600 m a.s.l. had rather constant Hg concentrations
ranging from 1.7 to 2.0 pg g$^{-1}$ while at higher altitude (from 700 to 1100 m a.s.l.) the concentration decreased in the range
between 1.4 to 0.8 pg g$^{-1}$. We found a linear relationship between altitude and mean Hg concentration with an R$^2$ of 0.78.
However, the concentration of Hg is not the correct parameter to evaluate the deposition flux from different sites since higher
snow accumulation might induce to a dilution effect. Therefore, the average Hg concentration was converted into Hg flux
considering the annual snowpack depth and its density. The estimated flux represent the total Hg load preserved in the annual
snowpack. The results suggest that the Hg deposition flux is similar at all the sites investigated (BRG 0.54; EDB 0.78; MLB
0.94; KNG 0.70; and HDF 0.58 µg m$^{-2}$ year$^{-1}$) and the elevation gradient is not statistically significant (R$^2$ = 0.28). In the snow
pit, the snow at the bottom is representative of the fall season, the middle of the winter deposition and the upper part
corresponds to the snow accumulated during the spring. The results show that the Hg concentration in all snow pits tend to
increase in the upper, most recent snow layer, in general from 0 to 40 cm depth. The upper part of the snowpack (the sampling
was done in April 2018) is more representative of the spring deposition, potentially influenced by Atmospheric mercury
depletion events (AMDE). More data analysis is required to verify this hypothesis.

### 3.4.3 Mercury in air of the Russian Arctic (Amderma)

Since June, 2001, a long-term monitoring of gaseous elemental Hg$^0$ in the troposphere was carried out near the Amderma
settlement (69.45° N, 61.39° E, 49 m a.s.l, Nenets Autonomous District, Russia), which is located on the Yugor Peninsula on
the shore of the Kara Sea and close to the Arctic border between Europe and Asia. A Tekran 2537A Hg$^0$ vapor analyzer, which
is a cold vapor atomic fluorescence spectrometer, was used. We carried out analysis of monthly, seasonal, and inter-annual
variability of concentration patterns, occurrence of both atmospheric Hg depletion (AMDE) and elevated (AMEE) events,
determined Hg fluxes to the atmosphere and evaluated concentration trends, and identified possible long-range transport cases
originating from Icelandic volcanic eruption with HYSPLIT - Hybrid Single-Particle Lagrangian Integrated Trajectory model
(Pankratov et al., 2015).

During the entire period of operation at this station, the location of the analyzer was changed three times, at different distances
of the Kara Sea coastline. The results show that the frequency of AMDE occurrence was shown to depend on the distance to
the coastline. From 2001 to 2004 the analyzer was located at a distance of about 9 km from the coast. The number of AMDEs
for this period was 10 % of the total number (6765) of measurements. During 2005–2010 period, the analyzer was placed at
2.5 km from the coast. As a result, the depletion events were registered more frequently, and especially during spring–summer
period. In 2006 and 2007 such depletion events were recorded in winter with a frequency of 20 % of the total number (1898)
of measurements. Starting from 2010 the analyzer was located at a distance of 200 m from the coast. As a result of this
relocation, more events were recorded (30 % of 67986 measurements) (Fig. 14a). For the first time in the Russian Arctic, we
identified the intensification of the AMDE frequency as a function of distance to the sea. The results are consistent with
observations in other polar stations such as Ny-Alesund and Andoya (Norway), Alert (Canada), and Pallas (Finland)
(Pankratov, 2015; Nguyen et al., 2009).



At Anderma, the probability density distribution of the Hg0 concentration was lognormal for the monitoring period from June
2010 to October 2013. There is a significant asymmetry in the left-hand region of the Hg0 concentration probability distribution
relative to the arithmetic average, pointing to the fact that low Hg0 concentrations are measured more frequently. In 2013 this
asymmetry was especially evident. The shift of the concentration to lower values was due to the increased amount of Hg0
depletion events recorded during the winter seasons of 2010–2013. To assess the dynamics of Hg, a linear approximation of
the average annual Hg concentrations for the lognormal distribution was calculated with the reliability coefficient R2 = 0.7
(Fig. 14b)

### 3.5 Halogens in the Arctic

Polar halogen activation occurs when halide ions originating from seawater are released via chemistry on snow/ice/aerosols to
the atmospheric boundary layer (e.g. Simpson et al., 2007; Abbatt et al. 2012). Bromide is a source of atmospheric bromine
($Br_2$), which is known to be an important species that controls the atmospheric ozone and Hg cycles in Arctic (see reviews of
Abbatt et al., 2012 and Simpson et al., 2015). Nearly complete ozone depletion events (ODEs) are regularly observed during
Arctic spring, when molecular bromine is released to the atmosphere through a complex multiphase process, which may require
the presence of sunlight and/or acids. Elevated concentrations of reactive halogens in the atmosphere are often co-observed
with atmospheric $Hg^0$ depletion events (Saiz-Lopez and von Glasow, 2012).

While observations are numerous, the ability of models to reproduce halogen activation and impacts has remained limited. For
example, WRF-Chem (Weather Research and Forecasting model coupled to Chemistry, Grell et al., 2005; Fast et al., 2006;
Peckham et al., 2011) is frequently used to study Arctic aerosol and trace gas cycling. Recently, WRF-Chem was improved
specifically for modeling ozone in the Arctic region (Marelle et al., 2017) although at present it does not include any description
of the impact of reactive halogens on the Arctic ozone cycle. Figure 15 shows the current ozone-predicting capability of the
model during spring and summer, compared to surface ozone observations at two Arctic sites. Within iCUPE, we aim to
improve model predictions of Arctic halogen activation and related ozone depletion in the atmosphere using the WRF-Chem
model. The ultimate goal is to improve the ability to predict the role of halogens in controlling the Arctic atmospheric mercury
cycle.

### 3.6 Persistent organic pollutants (POPs) and emerging contaminants (EOC) in the Arctic

Persistent organic pollutants (POPs) and newly emerging organic contaminants (EOCs) were analyzed from five Arctic
research stations, Alert (Canada), Pallas (Finland), Storhofdi (Iceland) and Zeppelin (Svalbard/Norway) since 1993 (Hung et
al., 2010) and at VRS (Greenland) since 2008 (Hung et al., 2016). Air concentration of polychlorinated biphenyls (PCBs),
organochlorine pesticides showed a more or less consistent decline in the 1990s. This reduction is, however, less apparent in
recent years. In contrast, concentrations of polybrominated diphenyl ethers (PBDEs), hexachlorobenzene (HCB) and some
PCB congeners were still found to be increased. The results indicate that both temporal and spatial pattern of POPs in Arctic
air is affected by anthropogenic emissions and may be affected by various processes driven by climate change, such as reduced
ice cover, increasing seawater temperature and increasing biomass burning in boreal regions.
EOCs such as poly- and perfluorinated alkyl substances (PFAS), novel brominated flame-retardants (BFR) and
organophosphate flame-retardants and plasticizers (OPFR) were investigated in the Arctic environment (Xie et al., 2015; Li et
al., 2017; Bossi et al., 2016). Long-range atmospheric processes may have moved particle-bound BFRs to the site, probably
during the Arctic haze season. Several modeling studies have been conducted in an attempt to resolve the dominant transport
pathway of PFASs to the Arctic, namely atmospheric transport of precursors versus direct transport via ocean currents (Wania,
2007; Armitage et al., 2009; Stemmler et al., 2010). Atmospheric measurements have shown the widespread occurrence of
PFAS precursors, e.g. fluorotelomer alcohol (FTOHs) and perfluorinated sulfonamide alcohols (FOSE/FOSA) (Shoeib et al.,



2006; Cai et al., 2012). The detection of perfluorocarboxylic acids (PFCAs), perfluorinated sulfonic acids (PFSAs) and neutral
PFAS in snow deposition is consistent with the volatile precursor transport hypothesis (Young et al., 2007; Xie et al., 2015).
The measurements of PFCAs and PFSAs in seawater from the Greenland Sea indicated melting snow and ice tend to be an
input source (Zhao et al., 2012). The inconsistent temporal and spatial trends between regions may be representative of
differences in emissions from source regions (Woehrnschimmel et al., 2013).

The decline of legacy POPs in the environment is expected in response to global efforts to reduce emissions. Whereas, with
their persistence in environmental matrices, such as water, sediment, soil, vegetation and ice/snow, warming Arctic may drive
them again into environmental circulation. Consequently, future research should be focused on quantifying remobilization
fluxes and sinks for both legacy POPs and emerging contaminants in the Arctic, and on developing a quantitative understanding
of global exposure pathways for POPs to support risk assessments (McLachlan et al. 2018).

### 3.7 Satellite remote sensing

Satellite remote sensing is a very useful tool for Earth observation in the Arctic, given the vast size of the area and constraints
in accessibility. The sensors that are typically applied are ranging from radar sensors of different frequency, typically X- and
C-band (TerraSAR-X, Tandem-X, Sentinel-1) to optical sensors (Landsat, MODIS, Sentinel-2 and -3) and altimeters (CryoSat-
2, ICESat-2, Sentinel-3). In future a hyperspectral mission (EnMAP) will survey the Arctic. In the framework of iCUPE, we
utilize satellite remote sensing data on the topics ranging from Arctic vegetation, snow-cover monitoring, and supraglacial
melt of glaciers. Beyond the development of products and assessment of existing approaches, also new techniques like
polarimetry for glaciers were assessed, which will serve for recommendations for future satellite missions. Furthermore, we
present an example for integration of satellite remote sensing, airborne observation and modelling, applied to a case of
anthropogenic emissions from oil/gas extraction in northern Russia. An analysis of the current observing systems and future
needs are summarized, including value chains of satellite missions.

### 3.7.1 Arctic vegetation

As part of iCUPE, we are working beyond the recent retrievals of satellite remote sensing data to include advanced current
and upcoming optical remote sensing missions with improved spatial, temporal and spectral resolutions and their potential for
characterization of Arctic regions. In particular, hyperspectral remote sensing (or imaging spectroscopy) has been shown to
provide superior derivation of key biophysical surface variables in snow-free permafrost areas during the summer months
based on field and airborne remote sensing data (Buchhorn et al., 2013; Bratsch et al., 2016; Liu et al., 2017; Beamish et al.,
2019a). The datasets compiled as part of the iCUPE project (see Sect 3.8) consist of four years of field data including canopy-
level spectral reflectance data, as well as aerial hyperspectral images (Alaska and Canada) and a dense Sentinel-2 time series
(Siberia). Additional ground-based data includes leaf-level photosynthetic pigment data from multiple phenological phases, as
well as aboveground biomass and detailed species composition data. The study sites include Toolik Lake Alaska, Qikiqtaruk-
Herschel Island (QHI) Canada, and Lena Delta, Siberia. All three sites represent key low Arctic research stations with the
Toolik and QHI sites having pre-established long-term vegetation monitoring plots. This dataset is unique and provides the
potential for further work given the freely available nature of much of the ancillary data and the intensive international research
being conducted at all locations. The main goal of this work was to provide an initial characterization and exploration of the
application of hyperspectral remote sensing data in Arctic terrestrial ecosystems where existing research is limited.

As a first step to improve the estimation and monitoring of Arctic tundra vegetation, we conducted a detailed spectral
characterization of dominant vegetation communities at Toolik Lake, Alaska with the aim of better informing current and
upcoming hyperspectral remote sensing platforms such as the PRISMA (Italy) mission (Loizzo et al., 2018) that started in
2019 and the EnMAP (Germany) satellite planned to be launched mid 2021 (Guanter et al., 2015). To be noted, as part of the
High Priority candidate missions for the Copernicus Sentinel expansion, the ESA is considering a hyperspectral mission named



CHIME for 'Copernicus Hyperspectral Imaging Mission for the Environment' (Rast et al., 2019). To accomplish the ground spectral characterization, we collected canopy-based spectral reflectance in five dominant low Arctic vegetation communities at three major phenological phases of leaf-out, maximum canopy, and senescence (Fig. 16) and simulated upcoming EnMAP and Sentinel-2 spectral reflectance. We then examined within and between community variability to determine the most discriminative wavelengths and phenological phase for the ground-based, EnMAP, and Sentinel-2 reflectance.

The results suggest that for imaging spectroscopy (ground-based and simulated EnMAP) the senescent phase imagery is superior to leaf-out and maximum canopy for differentiating vegetation types, while for Sentinel-2 maximum canopy was superior. The difference between the narrow and broadband data is likely due to the extreme color differences observed during senescence that are well captured by imaging spectroscopy but not by broadband data. These results provide important information for better interpreting current broadband and future narrowband spectral reflectance data for more accurate

estimation of vegetation composition, vigor and biomass (Beamish et al., 2017).

As a second step, we explored the relationship between spectral information from imaging spectroscopy and digital cameras data to biochemical variables at Toolik Lake, Alaska to assess the utility of low-tech cameras as a ground-based validation tool. To do this we looked at the relationships between narrowband and RGB indices and in turn the relationship of these indices to changes in photosynthetic pigments of chlorophyll and carotenoids. We were again interested in how these

relationships vary by phenological phase and vegetation type. The results showed that vegetation color contributes strongly to changes in both narrowband and RGB indices and that these changes are related to changes in photosynthetic pigment concentration (Fig. 17). We conclude that digital cameras can be used to estimate and track pigment development and degradation in low Arctic vegetation serving as a crucial in situ validation for hyperspectral remote sensing.

In a final effort to explore applications of hyperspectral remote sensing to characterization of Arctic vegetation, we explored

the spectral sensitivity of airborne hyperspectral imagery to fractional cover of plant functional types at Toolik Lake, Alaska (Fig. 18, Beamish et al., 2019b).

Using detailed plant compositional data, we explored the sensitivity using simple linear regression of individual wavelength and two-band spectral indices to varying fractional vegetation cover. We found that wavelengths and spectral indices showed overlapping areas of sensitivity to vascular and non-vascular functional types which could confound the extraction of

vegetation properties derived from vegetation indices. We also found that two-band vegetation indices do not provide consistent information across vegetation community types and the contribution of non-vascular and non-photosynthetic components influences indices values (Fig. 19). The findings of this research point to the importance of *a priori* knowledge of species composition and phenological phase to accurately interpret vegetation properties derived from imaging spectroscopy.

Using data from the Lena River Delta and Qikiqtaruk-Herschel Island, within iCUPE we plan to expand characterization of

Arctic vegetation using imaging spectroscopy to include biodiversity and biomass estimates. Using detailed species composition data and destructive biomass sampling in combination with spectral variables we will explore relationships at the ground, airborne and satellite scale to best estimate these key biophysical variables (Fig. 20).

In the next step, we aim to provide best practices and proxies for mapping vegetation status and aboveground biomass in the low Arctic at landscape scales with aerial imagery, Sentinel-2 and upcoming hyperspectral satellite imagery. The biomass data

will also be compared to volatile organic compound and atmospheric chemistry datasets.

### 3.7.2 Snow cover in the Arctic

Snow cover is an important component of the cryosphere that plays a key role for climate dynamics and the resources availability: the seasonality of the snow cover influences weather patterns, hydro-power generation, agriculture, forestry, tourism, and aquatic ecosystems (Beniston et al., 2018). The snow cover characterization and its annual spatial-evolution

represent important factors to be considered in the framework of climate modelling at a global scale. Furthermore, the snow cover has been officially declared as an Essential Climate Variable (ECV) by the Global Climate Observing System (GCOS)



and high priority is assigned to enhancing and maintaining snow cover observations (WMO, 2010; 2011). From this perspective, the continuous monitoring of snow cover is a major challenge of these last years and the advances in remote sensing explain why optical data are so diffusely used to monitor the snow covers.

The description of the snow cover implies two variables: the extent and the albedo. These variables, in the framework of the iCUPE project, are critical inputs for pollutant transport and climate change models. Remote sensing is the most common tool for the routine estimation of the snow cover extent, but two different aspects must be considered for the selection of the most appropriate input: time and spatial resolutions. Both components, using remotely sensed data, are connected to each other, since the higher the spatial resolution (below hundreds of meters), the lower the revisit time interval (more than 1 week) (Dietz

et al., 2012). Concerning the albedo, the spectral behavior of the snow reflectance in the visible and short-wave infrared ranges supports the discrimination of snowed surfaces from other matrices and, moreover, the characterization of the snow surface (Painter et al., 2003; Tedesco & Kokhanovsky, 2007). In details, the reflectance of pure snow in the visible range of the electromagnetic spectrum (400–700 nm) is approximately 100 % and it decreases as a function of the amount of impurity content in the snow and slightly increasing in the size of the snow grains with ageing. In the short-wave infrared part of the

solar spectrum (700–2500 nm), snow reflectance decreases rapidly and it is mostly controlled, in this case, by the snow grain size (Warren & Wiscombe, 1980; Wiscombe & Warren, 1980; Warren, 1982).

The state-of-the-art snow products concerning the snow extent are derived using remotely sensed data and they are based mainly on multispectral optical sensors. They can investigate the snow cover and give information about the size and the shape of snow grains (Dozier et al., 2009), the presence of impurity soot, the age of the snow, and the presence of depth hoar.

Furthermore, the short-wave infrared signal can support the discrimination between snow and clouds (Rodell et al., 2004). Considering that snow-covered surfaces are highly reflective in the visible range and low reflective in the Short-Wave InfraRed (SWIR) (Salomonson & Appel, 2006), it is possible to define an index that enhances the discrimination between snow and not snow in a single pixel. This index, defined as Normalized Difference Snow Index (NDSI), is calculated as follows:

$$NDSI = \frac{R_G - R_{SWIR}}{R_G + R_{SWIR}} \tag{1}$$


The green and the SWIR parameters are the bands available for each satellite sensor and their selection includes generally wavelength ranges centered at 500–600 nm ($R_G$) and 1500–1600 nm ($R_{swir}$). The estimation of the snow extent from remotely sensed multispectral images is based on the relation between the radiative behavior of the surfaces and the Fractional Snow Cover (FSC). This parameter describes the percentage of surface covered by snow (Painter et al., 2009) in a pixel element of

a remotely sensed image. The relation between the FSC and the NDSI represents the most common inference required by remote sensing studies. There are two options for estimating the NDSI—FSC relation: the first one consists in combining satellite products with different spatial resolution (Yin et al 2013); and the second one can be approached having a ground truth information. The first solution is based on (Salomonson & Appel, 2006) combining Landsat and MODIS data and a NDSI to FSC relation is defined as:


$$FSC = 1.45 \times NDSI - 0.01 \tag{2}$$

This knowledge was implemented in the SNOWMAP algorithm (Hall et al., 2016), which is the core of the MODIS data chain for the definition of remotely sensed snow products. This approach is replicated also on Landsat data (Vermote et al., 2016)

and in the NASA VIIRS ATBD (Riggs et al., 2015), derived by Suomi NPP satellite platform. Similarly, it is included in the SC algorithm, which is part of the Sentinel 2A data chain (Main-Korn et al., 2017). The second solution can be approached defining an empirical reflectance-to-snow-cover model that requires a calibration having a number of reference sites in the



satellite image (Solberg et al., 2006), which is the core of the GlobSnow Snow Extent (SE) data chain (Metsämäki et al., 2015). The occurrence of different interferences (cloud cover or snow metamorphism for example) requires the availability of
additional proxies, like webcam networks (Salzano et al., 2019), that can improve the calibration and validation processes of remotely sensed products.

Several satellite products are available for the remote sensing of the cryosphere and for this study we considered products obtained by optical sensors, characterized by different spatial resolutions: high (< 100 m); intermediate (100 m - 1 km); and low (> 1 km). The integration between those products and ground-based imagery will be tested, in order to improve the dataset
concerning the snow cover over a decade.

As part of iCUPE, we aim to develop a semi-automatic procedure focused on preparing a snow product useful for monitoring spatial and spectral variations of the snow cover. The first year was concerned on data collection and analysis of terrestrial images that will be complementary to Sentinel multispectral images. We selected a study area located close to Ny-Ålesund (Svalbard, Norway) where the Italian Arctic Station represents an important scientific facility and experimental activities are
appropriately supported. The site is characterized by intense international research collaborations and different data sources can be included as input for the integration between remotely sensed data and terrestrial images collected by webcam. We performed an overview about the algorithms focused on processing multispectral images and snow index calculation as well as on the automatic geo-rectification of ground-based photography and on the routinely estimation of the fractional snow cover. From this perspective, we carried out a first field campaign aimed to survey the available webcams in the study site and
to deploy a new webcam on the CNR Climate Change Tower. This site has proved to be the most suitable for the collection of ancillary data necessary for a correct interpretation of both satellite and terrestrial images. This CNR infrastructure is equipped with different meteorological sensors as well as instruments focused on detecting the radiative properties of the atmosphere and of the snow-covered surface.

The core of this activity was the development of a procedure suitable to collect remotely-sensed data and match these imagery
to FSC estimations obtained using terrestrial photography. The resampling of snow - no snow pixels, provided by webcams (Fig. 22 a and c), using grids associated with satellite imagery, provided a continuous dataset of FSC to NDSI relations (Fig. 22 b and d) that can enhance the recognition of different snow cover types in the considered site. From this point of view, terrestrial photography represents a term available daily or hourly in common for all the imagery that can be obtained by different satellites, with different revisit time (from 1 to 10 days) and under intense cloud cover (up to 80% of cloudy conditions
per year in Svalbard islands (Salzano et al., 2016). The final output will support the definition of the snow cover extent including shadowed areas and variations associated with the snow metamorphism. Furthermore, we will be able to describe the evolution of the snow cover year by year, with complete timelines that could assess the melt cycle variability over the last decades, on a local scale with terrestrial photography, or over the last few years, on a regional scale using remotely sensed data.

**3.7.3 Supraglacial lakes in Greenland**

The surface of ice sheets and glaciers in the Arctic, e.g. in Greenland, is prone to melt during summer. As part of iCUPE we focused on the Greenland ice sheet, given its current contribution to sea level rise and the polar amplification making this are to a hot spot of future warming. Whereas the entire ice sheet surface is experiencing melt only rarely, like in the extreme melt year 2012, the margins of the Greenland ice sheet is melting between May and August. Melt water may either percolate into
the porous firn matrix or may run-off along the surface and accumulate in topographic sinks forming supraglacial lakes. These supraglacial lakes may freeze over during winter, being covered with a lid of lake ice, or drain and deliver vast amounts of water acting as a lubricant to the glacier base. Whatever route the melt water takes, it also transports previously deposited pollutants during snow accumulation. There are three major questions that we addressed within iCUPE: (i) how much water





is drained? (ii) How often does the lake drainage occur? (iii) What are the opportunities and limitations of satellite radar remote
sensing to answer these questions?

Within iCUPE we studied an area in northeast Greenland, which contains lakes of the size of 21 km² and of up to 40 m depth, containing volumes of up to 108 m³ of water (Humbert et al., in review). The availability of water at the glacier base leads to an immediate speed of 9-22 %, that is declining over time as the water has been discharged (Neckel et al., in review). Estimating the volume of lakes has been done with differencing digital elevation models (DEM's) obtained from TanDEM-X at different
acquisition times. With revisit time of 11 days, we can limit the drainage duration with this no better than 11 days, however, the combination between optical (Sentinel-2) and TerraSAR-X data allowed to constrain the duration of one particular drainage event to be shorter than 1 day (Neckel et al., in review).

With Copernicus' Sentinel program, continuation of satellite missions is assured, which makes the development of methods to derive time series of lake drainage events based on these sensors a desirable approach for obtaining long time series in
future, as formation, filling and drainage of supraglacial lakes will increase in future under warming. Radar remote sensing is most attractive for observing polar areas, as it is independent of clouds and daylight. While the Copernicus program does not contain a bistatic radar mission required for lake volume estimates, Sentinel-1 is a C-band dual polarimetric mission, for which we explore the potential to retrieve quantities for the supraglacial hydrological cycle. Polarimetry allows to separate surface reflections from volume reflections, using transmission and retrieval of horizontal (H) and vertical (V) polarised radar waves
(Fig.23b,c).

The hypothesis we based our analysis on is that supraglacial lakes act as surface reflectors, whereas the surrounding glacier surface acts as a volume scattering material (Fig. 23a). With Sentinel-1 being only a dual polarimetric mission (HH, HV), the full decomposition cannot be conducted. However, the difference in amplitude between HH-HV has been used to derive a time series from May 2017 to September2019 over 79° N Glacier and Zacharias Isstrøm, two outlet glaciers of the North-East
Greenland Ice Stream. Figure 23 displays the difference HH-HV in April, thus before the onset of melt. Bright yellow color represents areas where surface scattering is dominating, whereas darker purple color represents volume scattering. The time series (not shown in Fig. 23) of difference in amplitude is prone to saturation during the melt period, consequently the difference HH-HV can also not be assessed. The areas with large differences HH-HV are coincident with lake locations detected with the DEM differencing, as well as optical satellite imagery, serving as a validation of this method. The vertical
polarization is arising from the reflection at the transition between lake ice and water. This is another indication for the lakes only being covered with a lid of lake ice and not freezing through entirely, which we also found in airborne radar surveys using AWI's ultra-wide band radar (UWB) on 2018-04-11.

We find polarimetry being a promising approach for detecting multi-annual lakes of large volume. Polarimetric SAR is likely to improve lake detection during the melt season and is currently investigated. From our assessment, it would be most
beneficial to have a full-polarimetric bistatic L-mission with short repeat cycles covering the polar areas, allowing even for SAR tomography, in order to be able to observe englacial aquifers and refrozen melt layers and their evolution in the next decades.

### 3.8 Integrating examples

This section presents a suite of integrative examples developed during the iCUPE project, which utilized the comprehensive
observation capacities and combined technologies in order to address satellite data validation task based on in situ observations, assessment of aerosol emissions from flaring in the Arctic in a combination of aircraft observations, satellite retrievals and modeling tools. We also present work towards development of proxy variables from the comprehensive observations and performed a value tree analysis for the Arctic observation network as a whole.



### 3.8.1 Satellite data validation

The SMEAR II station (Hari and Kulmala, 2005) in Hyytiälä is the ground-validation site for the current NASA Global Precipitation Mission (GPM, Skofronick-Jackson et al., 2018) and upcoming ESA EarthCARE satellites (Illingworth et al., 2015). The satellite calibration/validation activities can be divided into two main approaches, i.e. validation and verification of the assumptions and parameterizations used in the retrieval algorithms and direct validation of satellite observations.

Within iCUPE, the main focus of our satellite validation activities was on the microphysical assumptions and parameterizations

used in the ice cloud and snowfall retrievals. Given the climate and comprehensive surface-based observations of precipitating ice particles, our observations allowed us to constrain retrievals and refine these parametrizations. Following the work of von Lerber et al., (2017), the snowflake masses and size distribution were retrieved and used to derive event specific radar reflectivity-snowfall rate relations. These relations were applied in the validation of ice cloud and precipitation retrieval algorithms. For example, the performance of the "unified" algorithm that combines the radar and lidar observations of

EarthCARE, in a variety of snowfall conditions (Moisseev et al., 2017; Li et al., 2018), was tested using the snowfall data as shown in Mason et al. (2018; 2019). Using these studies showed that the measurements collected at Hyytiälä, namely combined observations of the surface snowfall observations and multi-frequency radar observations, are ideally suited for verification of satellite cloud and precipitation retrieval algorithms.

Direct validation of satellite precipitation products were carried out by applying tuned relations between radar reflectivity and

snowfall rate, as derived from surface precipitation measurements, to the Finnish Meteorological Institute Ikaalinen radar observations. This method was developed by von Lerber et al. (2018) and applied to validate GPM snowfall rate estimates at Hyytiälä, SMEAR II station. Using these observations, we showed that the current NASA GPM algorithms underestimates snowfall intensity, but showed good skill in detecting falling snow.

To prepare for the upcoming ESA Earthcare validation activities, preparatory studies were performed on how ground-based

radar observations can be used to validate space-based radar observations of clouds and precipitation. A comparison between such observations, in this case we use CloudSat, are shown in Fig. 24. As can be noticed there are detectable differences in the observed values. These differences are caused by attenuation of ground-based radar measurements in rain and melting layer. A method to address this challenge is described in (Li et al., 2019) and it will be used for the EarthCare studies in the future.

### 3.8.2 Anthropogenic emissions from oil/gas extraction over northern Russia: a combined approach using aircraft,
satellite data and modelling

Aircraft data collected for the first time over northern Russian oil and gas producing regions was used to examine the validity of available emission inventories. The data was collected in October 2014 and July 2017 as part of the French-Russian YAK-AEROSIB program (Paris et al., 2010; Antokhina et al., 2018). A combination of different approaches was applied involving joint analysis of aircraft and satellite data together with source-receptor modeling using FLEXPART/FLEXPART-WRF and

regional WRF-Chem modeling (Marelle et al., 2017, 2018) to investigate discrepancies in both black carbon (BC) and methane (CH$_4$) anthropogenic emissions over Russia, from where only very little data is available to date.

Analysis of Equivalent BC data collected over Northern Siberia in October 2014 showed clear signatures of enhanced concentrations in the lower troposphere especially near the surface (Antokhina et al., 2018). The origins of polluted air masses sampled during flights over the Ob Valley, Yamal and Kara Sea regions during October 2014 were analysed using WRF-Chem

BC tracer runs with emissions from different sectors using the ECLIPSEv5 (Klimont et al., 2017) and Arctic Black Carbon (ABC) (Huang et al., 2015) inventories with several plumes clearly identified as coming from the gas flaring regions. Backward FLEXPART-WRF simulations from the plumes were used to compare emission footprints for the Huang ABC and the ECLIPSE BC flaring emissions. Discrepancies were identified in both inventories and in some cases completely missing emissions, particularly over the Nenets region (Onishi et al., 2019, in prep.). The data is being further analysed using daily

VIIRS (Visible Infrared Imaging Radiometer Suite) satellite night-light data (provided courtesy of Earth Observation Group,



Payne Institute for Public Policy, https://eogdata.mines.edu/download_dnb_composites.html) to examine the sensitivity of modelled BC to daily variability in flaring emissions. Preliminary results suggest that the simulated BC is highly sensitive to significant observed variability.

Methane emissions from oil and gas activities are similarly poorly known, and their spatial and temporal distribution is related to that of BC (e.g. Gvakharia et al., 2017). Inventories are affected by 1) highly variable emission factors, 2) highly variable scheduled or unscheduled operations leading to release of methane and 3) the uncertain spatial distribution of regional totals (Fig. 26a). The inventory EDGAR v4.3.2 (Janssens-Maenhout et al., 2017) predicts a total emission for energy in Western Siberia of 2.8 MtCH4 yr⁻¹ whereas ECLIPSE v5a (Höglund-Isaksson et al., 2012) yields 14.1 MtCH4 yr⁻¹ (Fig. 26b). Similarly to BC, the YAK-AEROSIB campaigns were used to integrate in situ observations in methane inventory validation. A

comparison between observations and modelled methane enhancement using the FLEXPART model revealed overall that the most descriptive signals are contained in the occurrences of boundary layer measurements. Typically, the methane enhancements in these measurements are estimated by a tagged tracer analysis to originate for approximately 60 % from regional wetland emissions. We used the ORCHIDEE model (Poulter et al., 2017) to subtract the contribution of these widespread ecosystems to the methane enhancements detected in the boundary layer and estimate the compatibility of emission

inventories with the airborne measurements. However, this introduces a significant uncertainty in the constraint that can be applied to methane emissions from oil and gas that needs to be investigated though comparison with a different wetland model.

### 3.8.3. Aerosol vertical profiling in Russia and in Finland

Within iCUPE, data collected at ground-based lidar sites in Siberia and in Southern Finland provided new insights into aerosol sources affecting atmospheric composition in the Arctic and sub-Arctic regions.

**Russia**

Aerosol backscatter vertical profile data collected in Tomsk, Siberia and corresponding aerosol optical depth (AOD) have been derived using a micropulse lidar at 808 nm and a novel lidar data processing approach based on a careful system calibration and an aerosol source apportionment based on FLEXPART backward transport simulations of an aerosol tracer including dry and wet removal processes (Ancellet et al., 2019). The potential emission sensitivity maps from the FLEXPART simulations

were coupled with the spatial distribution of five aerosol sources to obtain the aerosol source apportionment: urban pollution map (cities of more than 500,000 inhabitants), biomass burning daily maps (NASA Fire Information for Resource Management System (FIRMS) using MODIS (Giglio et al., 2003) and the Visible Infrared Imaging Radiometer Suite (VIIRS) (Schroeder et al., 2014)), forest and desert maps (Advanced Very High Resolution Radiometer (AVHRR) database (Sertel et al., 2010)), and gas flaring emission map (ECLIPSEv54 database (Evaluating the CLimate and air quality ImPacts of Short-

livEd pollutants) described in Klimont et al., 2017).

An example of lidar aerosol vertical profile is shown in Fig. 27a for a mixture of gas flaring emissions from the Ob Valley below 3 km (Fig. 27e) and a dust layer transported from Kazakhstan above 3 km (Fig. 27c) during a time period with no forest fires in Siberia. Aerosol scattering ratios at 808 nm increased by a factor of 3 due to this event while the corresponding AOD reached 0.3 in good agreement with the Tomsk AERONET sun photometer observations and the MODIS hot spot near

TOMSK (Fig. 27b). Elevated CO columns seen by IASI above the Ob valley and Tomsk also support gas flaring emissions during this period. A CALIOP overpass (black line in Fig. 27b and d) also provides a latitudinal distribution on the aerosol layers showing the limited vertical extent (2 km) of the aerosol layer at 55° N transported from the gas flaring region, while dust layers reached 4 km at latitudes below 50° N over Kazakhstan.

The fraction of all the aerosol layers detected by the lidar from April 2015 to October 2016 were derived for the five aerosol

types observed in Ancellet et al. (2019). The results show that the occurrence of layers was linked to natural emissions (vegetation, forest fires and dust) and their contribution was high (56 %). However, anthropogenic emissions still contributed up to 44 % of the detected layers (1/3 from flaring and 2/3 from urban emissions). In Tomsk, the frequency of dust events is





very low (5 %). The contribution to the largest AOD (> 0.1) showed that the frequency of forest fires (25 %) and urban pollution events (25 %) were dominant, while the frequency of flaring (10 %) and dust emission (13 %) are equivalent and two times lower in frequency. The flaring emissions are indeed frequent but do not contribute very much to the large optical depth cases. Our results show that the aerosol particles originating from the urban and flaring emissions remain confined below 2.5 km, while the aerosol particles related to dust events were mainly observed above 2.5 km. The aerosols from forest fire emissions are on the opposite observed both within and above the PBL.

**Finland**

Based on a combination of ground-based High Spectral Resolution Lidar (HSRL, Shipley et al., 1983) observations and airborne in situ measurements, we have analyzed the boundary layer and aerosol layers aloft over a relatively clean SMEAR II station. An HSRL was one of the deployed instruments during BAECC (Biogenic Aerosols – Effects on Cloud and Climate) Campaign (Petäjä et al., 2016) that was organized at the station from February to September 2014. A Scanning Mobility Particle Sizer (SMPS) and Optical Particle Sizer (OPS) were installed onboard Cessna FR172F aircraft (Schobesberger et al., 2013, Leino et al. 2019) to measure aerosol size distribution from 0.1 to 0.23 $\mu$m and 0.3 to 5 $\mu$m, respectively. The flights were conducted in the vicinity of the station not higher than 4 km in altitude with relatively low speed of 200 km/h, thus, providing observations with good spatial resolution.

We analyzed a set of clear-sky and cloudy cases (Nikandrova et al., 2018) at Hyytiälä during the BAECC campaign. HYSPLIT 96 h backward trajectories were calculated every 50 m and combined into layers based on the similarities in the origin and traveling path, and compared with the layers recognized with HSRL. Most of the layers were recognized in both approaches yet arrival heights of the back trajectories were not always similar to the heights of the layers of HSRL. These discrepancies are due to the small-scale vertical motions that are not resolved in the HYSPLIT model (Stein et al., 2015).

In both clear-sky and cloudy cases, elevated layers with high aerosol concentrations were detected with both the HSRL and with the airborne measurements. One of the clear-sky case studies of 9 April 2014 is shown in figure 28. The boundary layer can be visually recognized from the backscatter coefficient to be around 1.4 km in the afternoon when the flight took place (Fig. 28a). Figure 28b shows HYSPLIT backward trajectories divided into three layers and their arrival heights. The aircraft data showed a similar shape of the size distribution for the elevated layers and in the boundary layer during the flight with higher aerosol concentration in the boundary layer (Fig. 28c). Small variability of the size distribution in the BL suggest a well-mixed BL, whereas in the elevated layers the internal variability was larger and the layers were not mixed as thoroughly.

Back trajectories analyses showed that the air mass origin was similar regardless of the arrival height, which indicated that both the BL and the elevated layers were affected by similar aerosol sources (Fig. 28d). Therefore, the differences in the number concentration and size distribution were mostly due to differences in their dilution during transport to Hyytiälä. Nucleation mode particles are also seen in the middle layer, and, assuming that this layer did not experience strong mixing for several days, we suppose that these new particles were formed in this elevated layer.

Certainly, higher aerosol concentrations within the elevated layers are periodically present in Finland. These layers can influence for example the columnar optical closure in the boreal environment (Zieger et al., 2015). More detailed analysis is needed in order to address the relative contribution of different aerosol sources to these layers and to be able to compare the results with the data available in Tomsk, Russia.

### 3.8.4 Deriving proxy variables from in-situ and satellite data

In iCUPE, we developed proxies for atmospheric variables from the in-situ data gathered in Finnish measurement sites. In practice, this means that we derive formulas that describe the proxy variables as functions of more commonly measured parameters. Previously, the proxies have been derived mainly for concentrations of gas phase molecules, such as sulfuric acid (Petäjä et al., 2009; Mikkonen et al., 2011) and monoterpenes (Kontkanen et al., 2016). In iCUPE, we are developing proxies for condensation sink (CS), boundary layer height (BLH) and ecosystem level gross primary production (GPP). The aim is to



produce proxies that rely on variables that can also be retrieved from the satellite data in order to produce proxies with good spatial coverage (e.g. Kulmala et al. 2011b). Here, we describe the CS proxy development, while the proxies for BLH and GPP remain under development.

Condensation sink (CS) describes the loss rate of vapor molecules due to their condensation on the aerosol particle surfaces (Kulmala et al., 2001). Therefore, it depends strongly on the particle surface area, which is determined mainly by the

concentration of particles with diameters close to or over 100 nm. Particles in this size range are introduced into atmosphere directly from various combustion sources, especially from residential combustion of wood and other biofuel (Paasonen et al., 2016), and grown from smaller sizes by condensation of mainly biogenic vapors (Riipinen et al., 2011; Paasonen et al., 2013). In our proxy we use carbon monoxide (CO) as a tracer for aerosols produced in combustion processes, and air temperature for estimating the formation of biogenic secondary organic aerosol. Air temperature has been shown to correlate with the

concentrations of the particles in the relevant size range in various continental environments (Paasonen et al., 2013).

We determined the proxy for daily averages of CS for measurement sites at SMEAR II in Hyytiälä (Southern Finland), at SMEAR I in Värriö and in Pallas (Northern Finland). When determining the dependence between CO concentration and CS, we observed a clear pattern in the seasonal minima of CO concentration at both sites (Fig. 29a). Since this pattern is not reflected in CS, but originates presumably from longer atmospheric lifetime of CO during dark months, we proceeded with

inspecting the relation between CS and ΔCO, which describes the difference between the observation and the 5[th] percentile of concentrations at the respected time of year. In figure 29b we show the dependence of CS on ΔCO in different temperature bins in Värriö. We calculated average slopes ($b_{ave}$) for the fittings in Fig. 29b for each measurement site, and evaluated the intercepts of the new fittings, where this average slope was applied. These intercepts are depicted for the sites in Fig. 29c. These intercepts reflect the expected temperature dependence, where CS is not impacted by temperature at T < 5°C, when

ecosystem emissions are generally low, and increasing positive temperatures cause increase in CS. Finally, we formulated the temperature dependent part of the proxy ($a_1(T)$) by calculating the average intercepts in T < 5°C and making fit to the intercepts in positive temperatures (Fig. 29c). This produces the final proxies for CS (Fig. 29d), with form:

$$CS_{proxy} = a_1(T) + b_{ave} \times \Delta CO.$$

As a next step for the CS proxies, we will derive and test a proxy derived from a data set combined from all different sites,

study how the proxies work for data from other polar stations, and investigate whether CO and temperature retrieved from the satellite data can be applied for the proxies.

### 3.8.5 iCUPE data flow

One of the main outcomes of iCUPE are new data products developed based on the comprehensive use of in situ and satellite remote sensing. The new iCUPE data products, which will fill e.g. observational gap of the key variables of Persisting Organic

Pollutants (POPs), Chemicals of Emerging Concern (CECs), Short-Lived Climate Forcers (SLCFs), and atmospheric trace gases in the polar context, are being co-designed together with different end users. The list of data products is presented in Table 1. End-users for the iCUPE data sets include European Environment Agency, geoengineering bodies, decision makers, intergovernmental organizations, local government and environmental administration (ministries), citizens and National and European weather services. The data can be used to improve models, to advance our scientific understanding of the Arctic

regions, support policy making and improve weather predictions and mechanisms in risk control of natural hazards, in sea traffic control, in tourism and in chemical weather prediction.

iCUPE will provide in total more than 20 datasets as products usable for researchers, decision- and policy makers, stakeholders and end-users communities. These will be a valuable and important contribution from the iCUPE project activities in piloting open access to Arctic data. All these datasets will be publicly available for different applications and services. Focusing on the



Arctic region territories, the planned datasets will include e.g. novel data on anthropogenic contaminants in snow and ice cores and organic contaminants in the air-snow-water and concentrations of different chemical species and aerosols as well as their characteristics including vertical profiles. Within iCUPE we will also develop specific datasets focusing on selected geographical areas in northern latitudes. The data provision is open and Russian iCUPE collaborators have also provided data from the Russian Arctic, such as atmospheric mercury measurements at Amderma station, elemental and organic carbon over

the north-western coast of the Kandalaksha Bay of the White Sea and micro-climatic features and Urban Heat Island intensity in cities of Arctic region.

   Each of the data sets are promoted through "teasers" ([www.atm.helsinki.fi/icupe/index.php/submitted-datasets](www.atm.helsinki.fi/icupe/index.php/submitted-datasets)), which provides the metadata information pertinent to the data sets. Majority of archived datasets (as products) are directly linked (and downloadable) and corresponding metadata information included. The University of Helsinki will take responsibility for

long-term storage, accessibility, and maintenance. The raw data will be hosted and maintained by the dataset providers.

   Selected iCUPE datasets are also to be tested and integrated into several platforms. To facilitate and standardize access to data cloud-based online platforms, known as the Data and Information Access Services (DIAS), providing storage, centralized access and handling to data, and processing tools. The DIAS platforms (CREODIAS - creodias.eu; SOBLOO - sobloo.eu; MUNDI - mundiwebservices.com; ONDA - www.onda-dias.eu/cms; and WEkEO - www.wekeo.eu) allow users to explore,

process, and download Copernicus data and information as well as to have the ability to process and combine with data from other sources. It is also possible to develop and host new applications on such platforms. Other tested platforms for pre/post-processing/analysis of iCUPE data include the Virtual Laboratory ([vlab.geodab.org](vlab.geodab.org)) Google Earth Engine ([earthengine.google.com](earthengine.google.com)), Polar Thematic Exploitation Platform ([portal.polartep.io](portal.polartep.io)), Global Earth Observation System of Systems Portal ([www.geoportal.org](www.geoportal.org)). We will continue to explore these options for the iCUPE data provision in the future.

All final data products and proxies developed will follow interoperability and data sharing principles endorsed by GEOSS. We explore possibilities to maximize the use of new data products and advertising these to larger communities, such as in several community portals such as Sustaining Arctic Observation Networks (SAON) Arctic Observation System and Arctic Portal and Pan Eurasian Experiment (PEEX) Program- the Arctic-Boreal Hub Portal. The SAON portal expands the visibility of the iCUPE data to the circumpolar context and PEEX to Russian and Chinese research and other end-user communities.

For selected topical data sets, some will be included in the University of Helsinki smart-SMEAR platform (Junninen et al. 2009) and disseminated via metadata catalogs to a wider audience. As a whole we need to utilize the full capacity of the multi-platform approach (in situ, satellites and models) to address the fate of pollutants in the Arctic. This will provide tools to perform targeted reductions, but can also provide new insights into the lifecycle of the pollutants in the Arctic environment.

### 3.8.6 Observation system analysis

In addition to investigate new observation capabilities, iCUPE was tasked to evaluate the connectivity and scale of satellite observations for Arctic needs. This task was expanded and elaborated by the SAON initiative "Arctic observing system assessment framework" (STPI/SAON, 2017). The framework links societal benefits to the observing system. Within the framework we identified 170 common objectives classified under 12 Societal Benefit Areas from international Arctic strategy documents. These objectives identify a need for earth observation information to enable informed decision-making in the

Arctic context. In this work, a team of domain experts from SAON, AMAP and iCUPE evaluated the assessment framework value tree for atmosphere and ocean variables.

   The basis of the analysis includes the observation capacity. Earth Observation (EO) inputs like SYNOP station measurements of physical atmosphere and in other stations, ocean variables, were linked to key products / outcomes / services such as numerical weather prediction and through groups like in this case weather service connected to key objectives of the assessment

framework (Fig. 30), such as weather service or environmental information service.



We weighted the analysis by annual costs for operation. Representative yearly unit costs of EO inputs and modelling components were estimated by station experts or estimated based on European Union projects or Copernicus program tenders. As the observational capacity and its monetary value, we utilized the WMO OSCAR database for satellite and surface observation systems (https://www.wmo-sat.info/oscar/) north of 60° N. This provided us the volume of stations in different

station and mission categories in the Arctic (Table 2). Based on our analysis, the total yearly value of this observation system including EO inputs and modeling is over 204 million €. Compared to the observing system estimated costs in the area 30° N to 60° N this is only about one fifth.

Another way to scale Arctic needs is to look at the distribution of stations globally for operational networks: WMO has organized in the integrated ground observing system WIGOS stations on land, co-sponsoring is mainly with IMO for marine

platforms and the non-affiliated are research organizations distributing information in near-real-time. The Arctic slice compared to Northern mid-latitudes has about one sixth of the stations on about one third of aerial coverage compared with the mid-latitude costs. Doubling current observing efforts would be needed to achieve a similar coverage in the Arctic.

The value chain from the observations through modeling frameworks into services and decision-making is presented in Fig. 30. It is an estimation of the data flows from different stations and satellites to modeling systems and onwards to services

carrying the weight of the total investment forward divided on the importance that the next step places on the information. The costs of modeling have been estimated and added to the value tree. Services are an important element, included by giving each connection in the tree a value of 1 million annually. This cost represents the global effort and with the included connections amounts to 81 million EUR per year. To compare this value, it corresponds to the annual costs of the Finnish Meteorological Institute at 74 million €, which operates 429 meteorological stations, all included as part of Arctic observation capacity in

Table 2. In addition to FMI, there is at least 6 similar other institutes serving for the Arctic, the estimate is reasonable.

The decision-making step in the value tree is channeled through 6 services, which include research activities and the efforts to consolidate research results in Arctic Council working group reports or the International Panel on Climate Change or the IPBES for biodiversity as one-off actions. Continuous services are established in weather, climate, marine and environmental information. The full value tree that connects all services to 170 key objectives from the Societal Benefit Areas is complex,

but could help to attribute the costs of an observing system fairly to all the areas that use it. Browsing the full tree is there much better on the web under http://arctic-obs.fmi.fi/ and in the publication (Strahlendorff et al., 2019). iCUPE activities are grouped in the value tree under research stations, observation grids and research services. The costs could not be determined in detail, but the activities fit into the research station estimate although the in situ monitoring activities are very diverse. Travel costs can be substantial, but this component could not be averaged usefully over the range of different research stations.

In conclusion a doubling of Arctic observing efforts both from satellites as well as in situ is necessary and reasoned. This amounts to a yearly increase of 200 million € to observing infrastructure and the production value chain. The European Union Space program is planning for additions to the satellite component with Sentinel Expansion missions, but additions to the in situ component are missing a funding action.

**4 Summary and Outlook**

The polar regions are facing changes in the future, both in terms of climate change influenced by Arctic amplification but also due to on-going megatrends such as globalization that drive development e.g. of new transport routes through the fragile Arctic environment and extensive utilization of natural resources within the Arctic. The environment in the Arctic is at risk. To address aspects related to the Arctic change, we set up the ERA-PLANET Strand 4 project "iCUPE - integrative and Comprehensive Understanding on Polar Environments" to provide novel insights and observational data on global grand

challenges with a polar focus.


In iCUPE, our concept is to utilize an integrated approach combining in situ observations, satellite remote sensing Earth Observations (EO) and multi-scale modeling to synthesize data from comprehensive long-term measurements, intensive campaigns and satellites to deliver data products, metrics and indicators to the stakeholders concerning the environmental status, availability and extraction of natural resources in the polar areas.

The circumpolar coverage of in situ atmospheric observations has developed during the last decades. In selected locations, such as Svalbard, observations are available for assessing the decadal variability of key observables, such as black carbon. The geographical coverage should be particularly expanded in the Russian Arctic (Petäjä et al., 2019) although a lot of observation sites are operated already (Alekseychik et al., 2016). However, the connection of various continuous national activities and intensive campaigns within the Arctic should be coordinated in order to reduce operation costs. The observations should take

advantage of data harmonization procedures set up e.g. by WMO-GAW and European Research infrastructures (ICOS, ACTRIS, eLTER). Work towards open data sharing should be continued.

Harmonized and open data is crucial for development of services based on the observations within the Arctic. We need to maintain and improve comprehensive and continuous observations network of in situ observations in the Arctic that is sustained for extended periods of time to monitor the concentration of atmospheric pollutants. This gives us verification data

on the political decisions to reduce the emissions of harmful compounds but also enables us to respond to arising new threats to the environment. The in situ observations analyzed and performed within iCUPE underlined that we need to utilize the full capacity of the multi-platform approach (in situ, satellites and models) to address the fate of pollutants in the Arctic. New observational needs should be analyzed e.g. in the framework of Essential Variables (EVs) that connect to sustainable development goals of the United Nations.

The long-term observations at Svalbard revealed the aerosol concentrations are strongly modulated with seasons. This is apparent in Svalbard Equivalent Black Carbon concentrations as well as in aerosol number concentrations detected at Villum Station in Northern Greenland. The comprehensive source apportionment of the organic aerosol fraction at different observation sites in the Arctic indicated that the concentrations are affected by a combination of regionally specific sources and long-range transport of anthropogenic aerosol particles. Vertical profiling can provide novel insights into pollution

transport and dispersion in the Arctic environment. The combined use of aircraft observations, modeling tools and emission inventory analysis can provide novel insights into variability of atmospheric pollutants within the Arctic.

Similar to the decrease in black carbon, the decline of legacy POPs in the environment is expected under global efforts. Whereas, with their persistence in environmental matrices, such as water, sediment, soil, vegetation and ice/snow, warming Arctic may drive them again available for environmental circulation. Consequently, future research should be focused on

quantifying these remobilization fluxes and sinks for both legacy POPs and emerging contaminants in the Arctic.

Within iCUPE, we determined concentrations of mercury in different compartments and fluxes between these compartments in the land-atmosphere continuum. The mercury flux to the Arctic environment through riverine discharge was found to be important. The concentration of mercury in the snow varied as a function of snow depth indicating a crucial role of atmospheric deposition. Long-term observations of atmospheric mercury in the Russian Arctic were found to be connected with the distance

to the sea. The modeling results underline the interconnected life-cycle between mercury halogen compounds and atmospheric oxidants.

Furthermore, the remote sensing activities in the iCUPE project highlighted the need of a ground truth concerning the fractional snow cover. This kind of information can be obtained by terrestrial photography and it represents a tool in common between remotely-sensed products useful for integrating satellite data with different spatial and time resolutions. This approach can be

very effective in high-latitude areas where illuminating conditions, topography and cloud cover limit the use of optical remote sensing.

The satellite remote sensing work within iCUPE revealed that imaging spectroscopy also known as hyperspectral remote sensing for Arctic vegetation provides valuable information on vegetation status and biomass that can improve our



understanding of long-term vegetation trends derived from broadband data. Spectral information stored in the narrow
wavelengths of the visible spectrum are the most promising for differentiating spectrally similar vegetation communities and
for delivering information on photosynthetic activity and biomass. The lack of ground-validated datasets and an overall scarcity
of dense, high quality image time series remains a challenge. The increasing availability of hyperspectral data will face similar
challenges and given the limited number of hyperspectral remote sensing studies in the Arctic, high quality, ground-validated
data is required to accurately interpret these dense and complex datasets. The rise of more advanced classification methods
such as machine-learning techniques are highly promising for Arctic vegetation mapping using advanced remote sensing
platforms. An increased effort to develop Arctic-specific algorithms is needed.

Within iCUPE, the benefits of combining in-situ and satellite remote sensing and multiscale modeling in cryospheric, terrestrial
and marine regimes as well as atmospheric domain are clear. Such integrative activities need to be expanded and continued in
order to provide verified environmental data and services in the changing Arctic.

**Acknowledgements**

This project has received funding from the European Union's Horizon 2020 research and innovation programme under grant
agreement No 689443 via project iCUPE (Integrative and Comprehensive Understanding on Polar Environments)" and through
ACTRIS2 is gratefully acknowledged. Additional financial support through Academy of Finland (Center of Excellence in
Atmospheric Sciences) and through Academy of Finland project NANOBIOMASS are gratefully acknowledged. P.O. thanks
for partial support from project Russian Foundation for Basic Research (RFBR) No. 18-05-60084. Part of this work was
supported in part by Ministry of Science of Education of RF (Agreement No. 14.616.21.0104, unique identifier
RFMEFI61618X0104). We thank J. Pelon for assistance with research conducted at LATMOS and IGE. French groups also
acknowledge support from the French Arctic Initiative project Pollution in the Arctic System (PARCS). Part of this study was
founded by the Danish Environmental Protection Agency and the Danish Energy Agency by means of DANCEA funds for
Environmental Support to the Arctic Region. The Royal Danish Air Force is acknowledged for providing free transport of
equipment to Station Nord, and the staff at Station Nord is especially acknowledged for excellent support. Part of this study
was funded by the German Science Foundation (Deutsche Forschungsgemeinschaft, DFG) as project with the number LA
2907/5-3, BA 1988/14-3, WI 1449/22-3. For UAV measurements in Ny-Ålesund we thank the whole AWIPEV team and
Kingsbay crew for support during preparation and for hosting us.

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

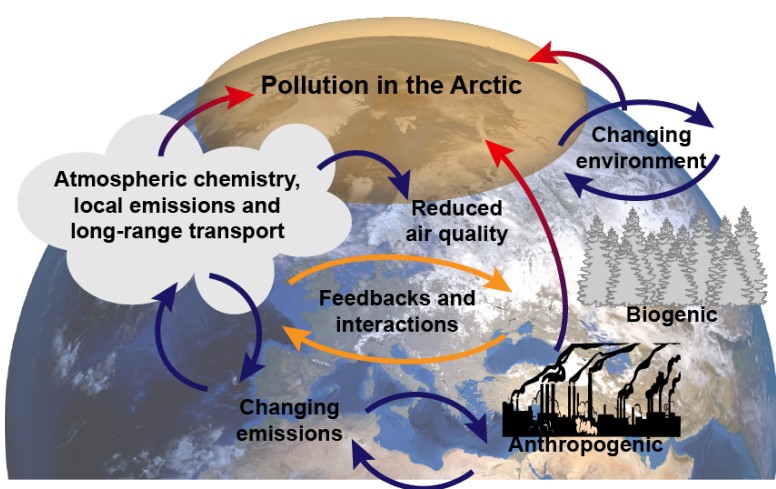

**Figure 1.** Atmospheric concentration of pollutants and their lifecycle in high latitudes are affected by local and regional anthropogenic activities and long-range transport from lower latitudes. Pollutant distributions and life cycles are modulated by transport patterns, changes in the biosphere, increased natural resource extraction and increased shipping in the Arctic Sea. Various feedbacks and interactions can either speed up or hinder the changes.




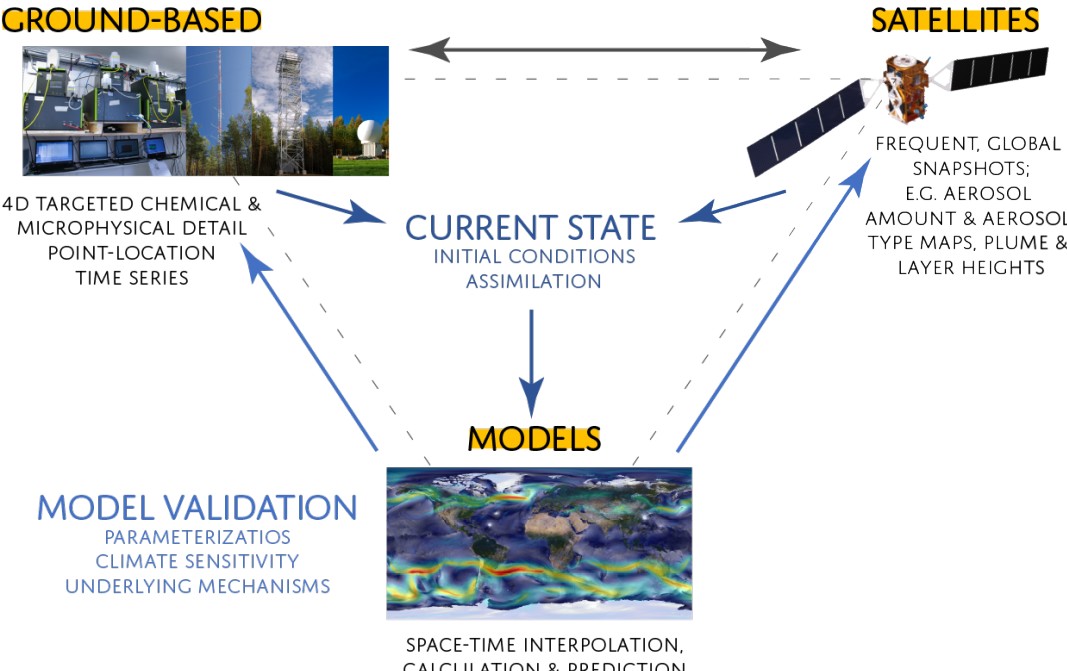

**Figure 2.** The integrative concept of iCUPE incorporates data and knowledge from ground-based observations, satellite remote sensing and modelling results providing a comprehensive view about the state of the environment in the polar areas.

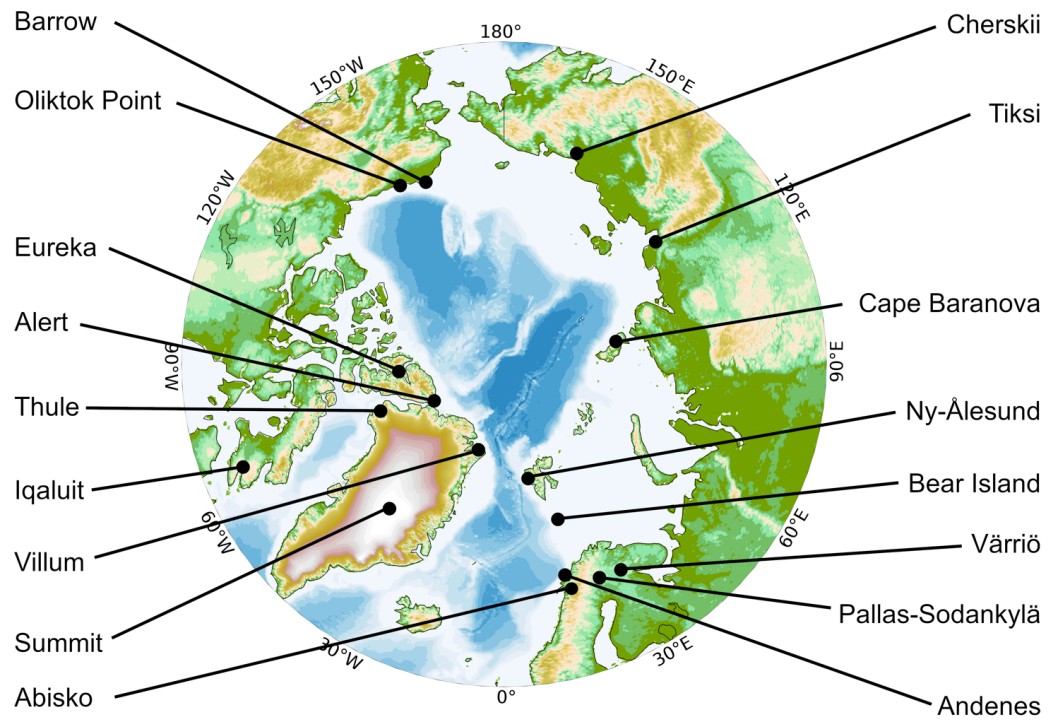

**Figure 3.** A map of stations with year-round observations in the Arctic with atmospheric aerosol measurements.





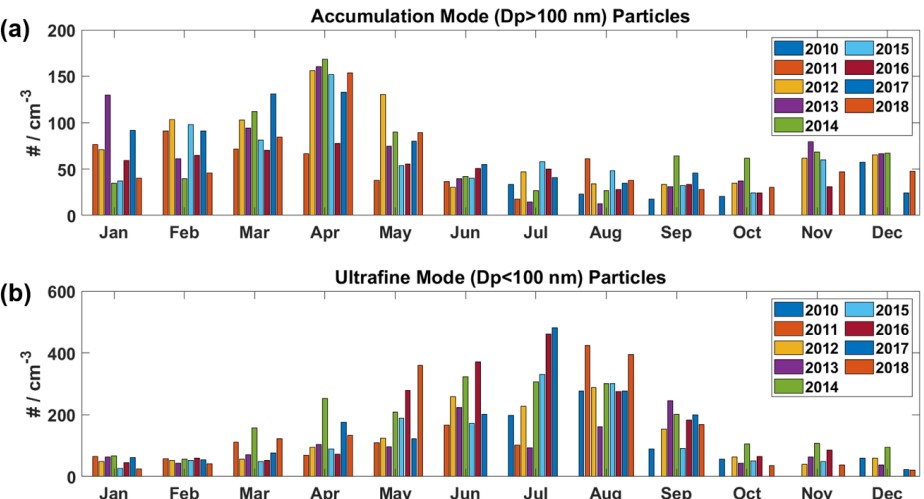


**Figure 4.** Monthly mean accumulation mode **(a)** and ultrafine mode **(b)** number concentration (# / cm⁻³) measured at VRS from 2010 to 2018. Note the difference in scale of the y-axis for each panel.

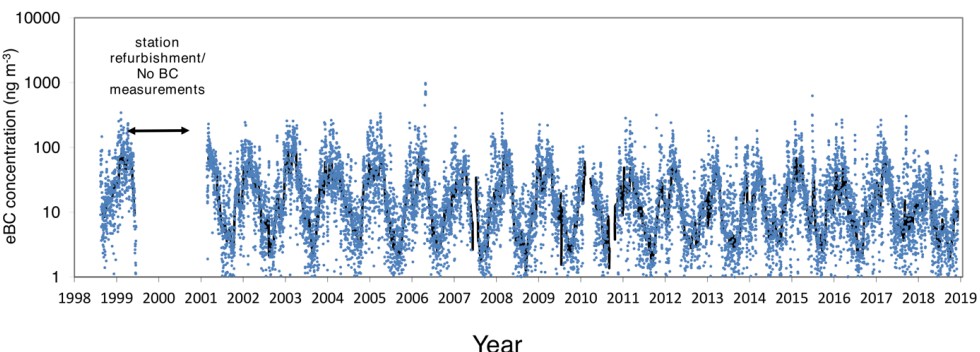

**Figure 5.** Equivalent Black Carbon concentration time series obtained by a 7-wavelength aethalometer at Zeppelin station.

(Data before 2000 are obtained by an AE-9 aethalometer)





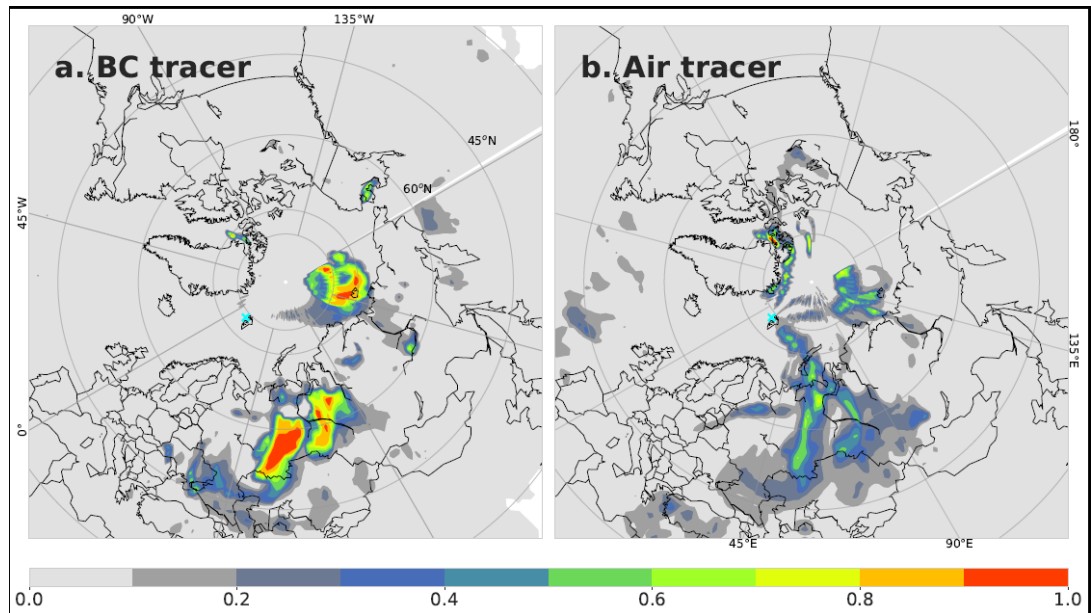

**Figure 6** Potential source contribution areas for BC observations at Zeppelin station Svalbard. The color scale represents statistical potential (1:100%) for 90th percentile of observed values to originate from certain areas of the Northern hemisphere. **(a)** and **(b)** corresponds to the air mass transport calculations using FLEXPART-model with the BC tracer and Air tracer respectively.


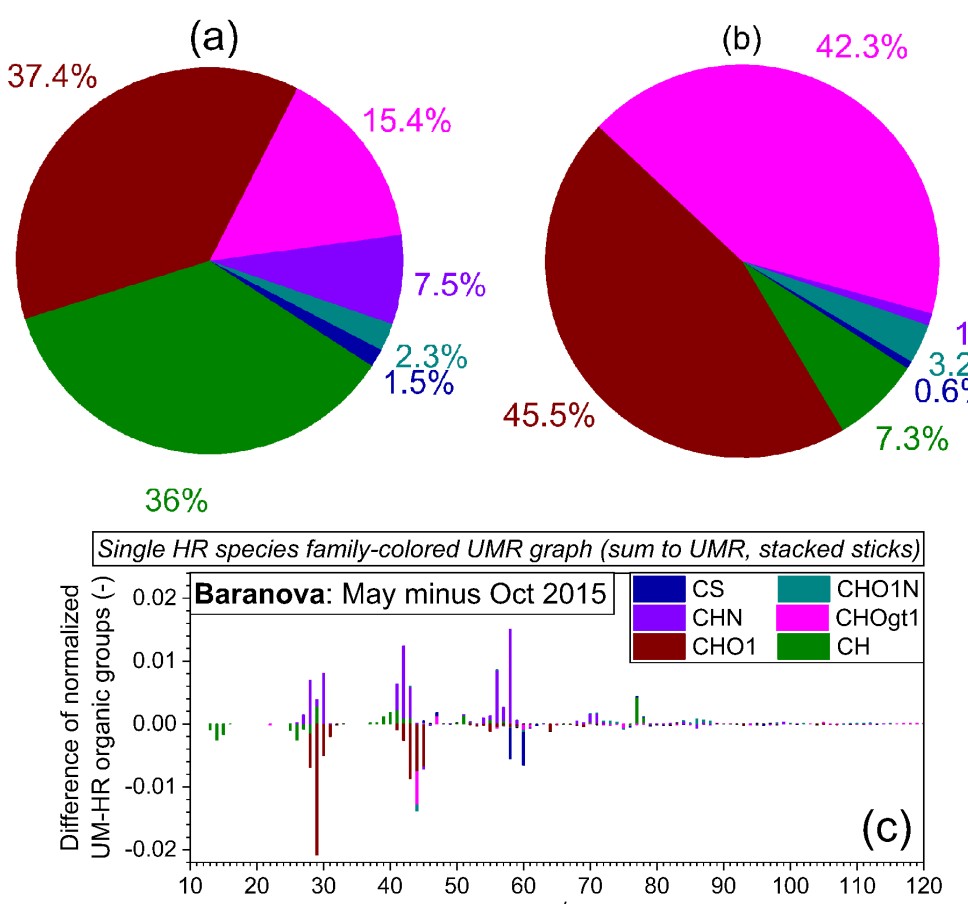

**Figure 7.** Offline AMS-based fractional contribution of the different organic aerosol families for **(a)** Cape Baranova (H:C = 1.53, O:C = 0.53) and **(b)** Alert (H:C = 0.81, O:C = 1.64) during October 2015. **(c)** Single organic species family-colored (stacked sticks) difference of the May minus October 2015 normalized mass spectrum. The former sample is expected to be influenced by pollution transport from mainland Russia. Data were taken and analyzed at high resolution, but are summed to unit mass resolution for display.

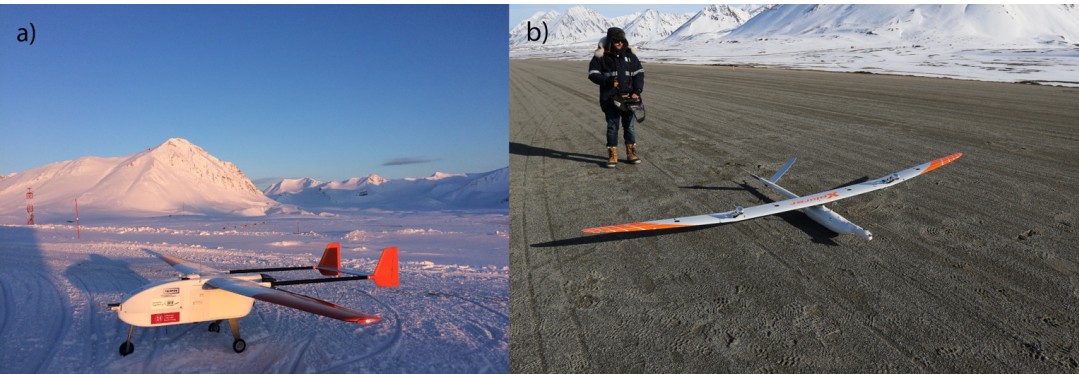

**Figure 8.** **(a)** The UAS ALADINA, **(b)** the UAS MASC3, both at the airport in Ny-Ålesund.

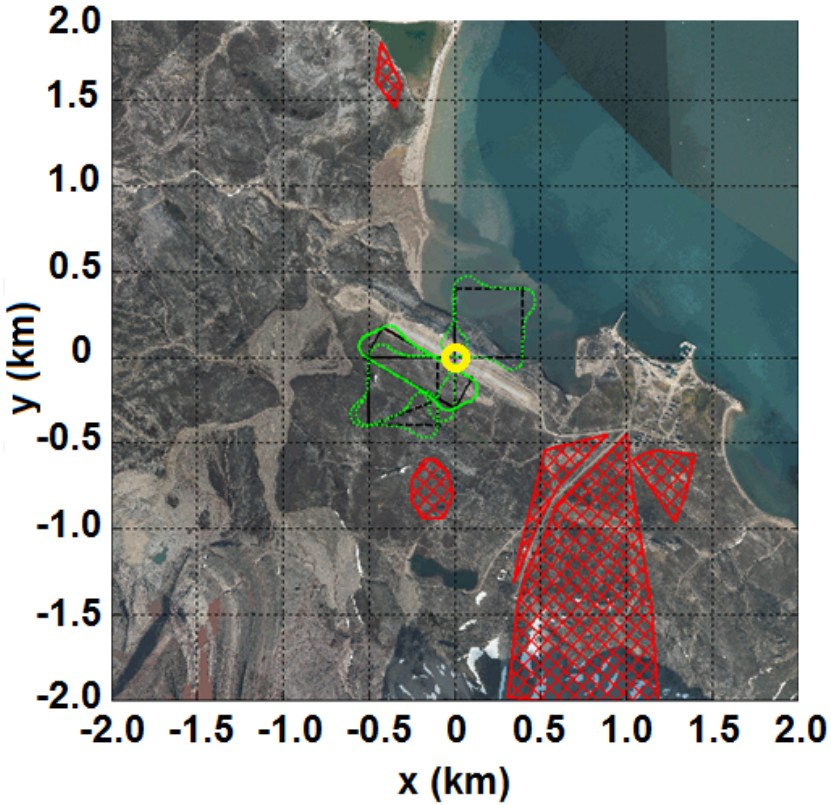

**Figure 9.** Flight path of the UAS ALADINA (start point in the centre, marked in yellow) in order to study the vertical and horizontal distribution of aerosol particles above ice surfaces near glacier and over open water. The TopoSvalbard map is obtained from Norwegian Polar Institute, retrieved from http://toposvalbard.npolar.no. ©



Figure 10. Selected profiles measured on May 23, 2018 in Ny-Ålesund for (a) 10:22 UTC and (b) 11:17 UTC. The panels
show vertical profiles of potential temperature, water vapor mixing ratio and aerosol particle number concentrations between
4 and 12 nm and above 390 nm.



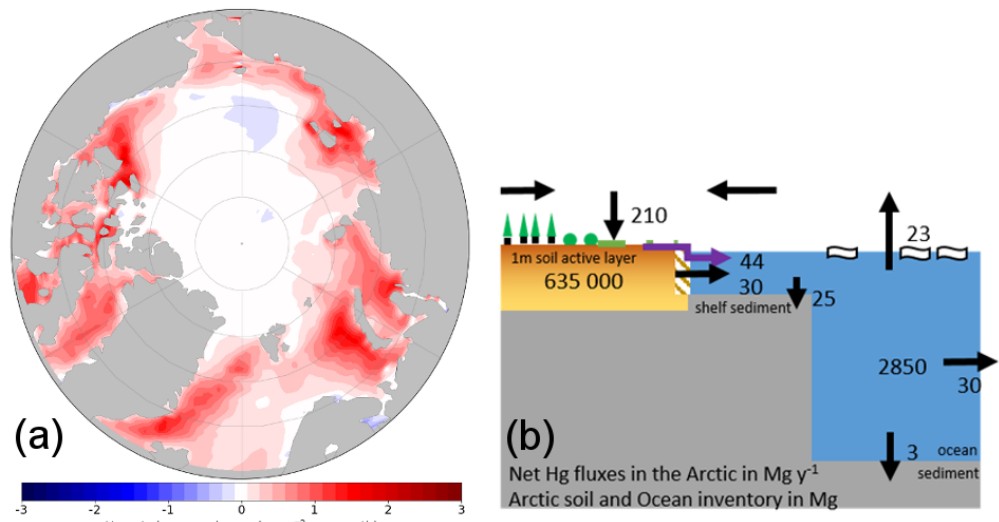

**Figure 11. (a)** Modeled net air–sea exchange of Hg ($\mu g \cdot m^{-2} \cdot month^{-1}$) across the Arctic Ocean basin for the months June–July from the coupled GEOS-Chem/MITgcm Hg chemistry and transport model. Positive numbers (red) indicate a flux to the atmosphere. **(b)** The modern Arctic Hg cycle, showing net fluxes (metric tons per year) between the different terrestrial, marine and atmospheric reservoirs (metric tons). The hatched area represents the coastal erosion Hg flux. The large river Hg flux (purple arrow) confirms a new paradigm where tundra vegetation and soil uptake mid-latitude Hg atmospheric emissions and transfer them to rivers and the Arctic Ocean. Reproduced from Sonke et al., 2018.

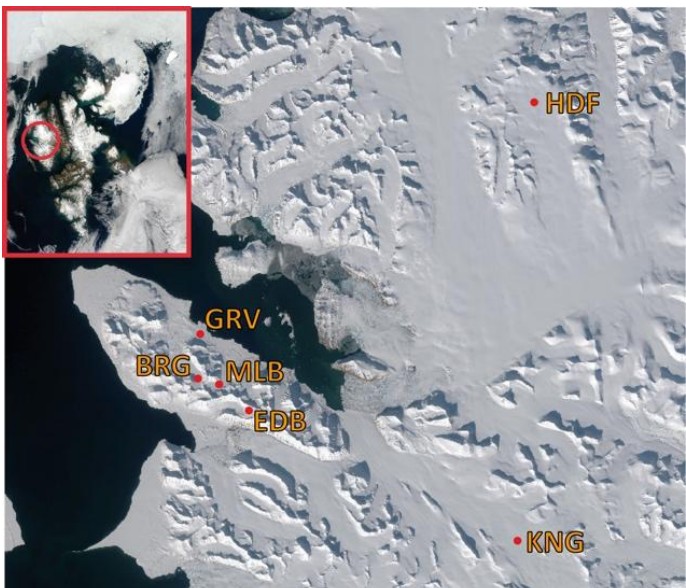

**Figure 12.** Snow sampling locations in the Svalbard Archipelago. BRG - Austre Broggerbreen glacier; MLB - Midtre Lovenbreen glacier; EDB - Edithbreen glacier; KNG - Kongsvegen; HDF - Holthedalfonna ice field. The TopoSvalbard map is obtained from ©Norwegian Polar Institute, retrieved from http://toposvalbard.npolar.no.



**Figure 13.** The upper panel (a) shows the elevation of the snow pit sampling sites. The lower panel (c) depicts Hg concentrations in the annual snowpack expressed in pg g⁻¹. The depth scale is expressed in cm and the snow surface is set at 0 cm. Samples have a constant resolution of 5 cm to the bottom. Middle panel (b): mean Hg concentration (color square) and total flux (color triangles), vertical bars indicate one standard deviation.





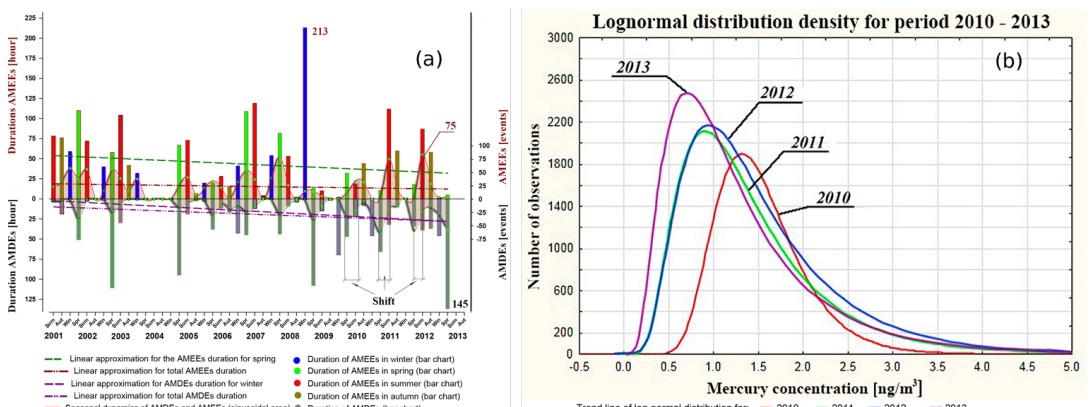

**Figure 14**. **(a)** Seasonal dynamics of AMDEs and AMEEs during the monitoring period from 2001 to 2013 (sinusoidal area). Duration of AMDEs and AMEEs (bar chart, the red color - summer, green - spring, dark yellow - autumn, blue - winter). Linear approximation for the AMEEs duration for spring (green dashed line), and AMDEs duration for winter (purple dashed line); for total AMEEs duration (brown dash with two dots) and for total AMDEs duration (purple dash with two dots). **(b)** The lognormal distribution of the atmospheric $Hg^0$ concentration during the monitoring period from June 2010 to October 2013.

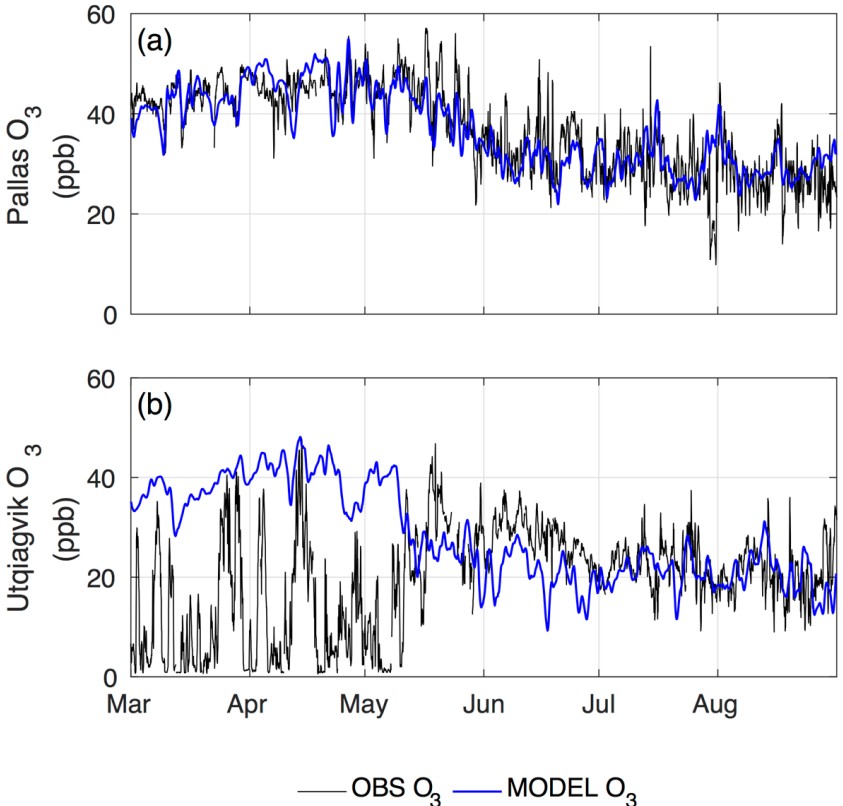

**Figure 15.** Observed and model predicted ozone concentrations at two Arctic surface stations for March–August 2012. WRF-Chem in blue and observations are in black. The model does not capture ozone depletion events observed at Utqiaġvik during Arctic spring. At Pallas, an inland site where no bromine activation is expected, the model in the present state captures the full spring-summer seasonal cycle of ozone.






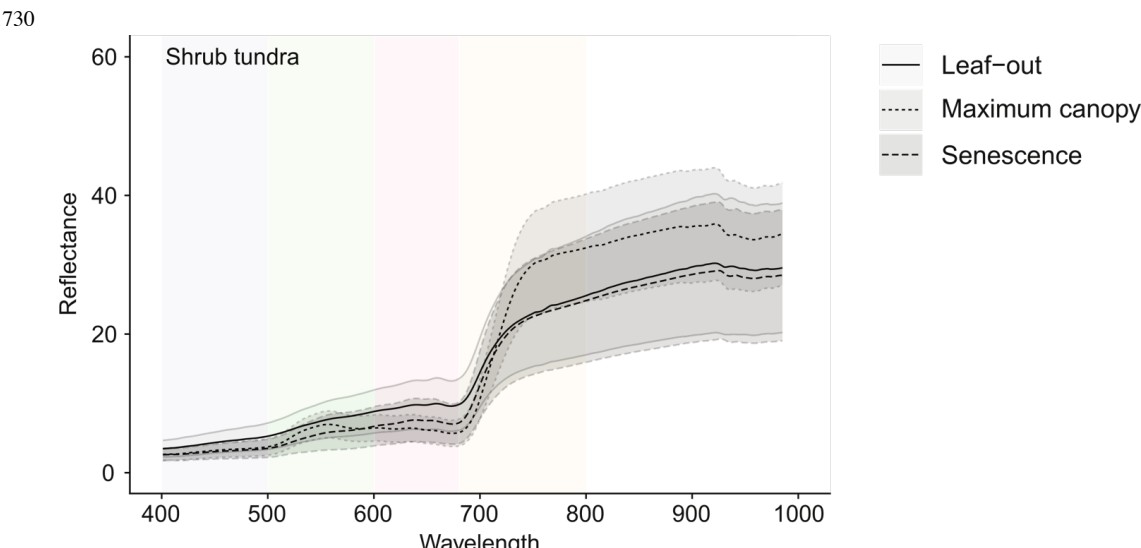

**Figure 16.** Canopy-level spectral reflectance from a dwarf shrub community at three major phenological phases of leaf-out, maximum canopy, and senescence (Beamish et al., 2017).

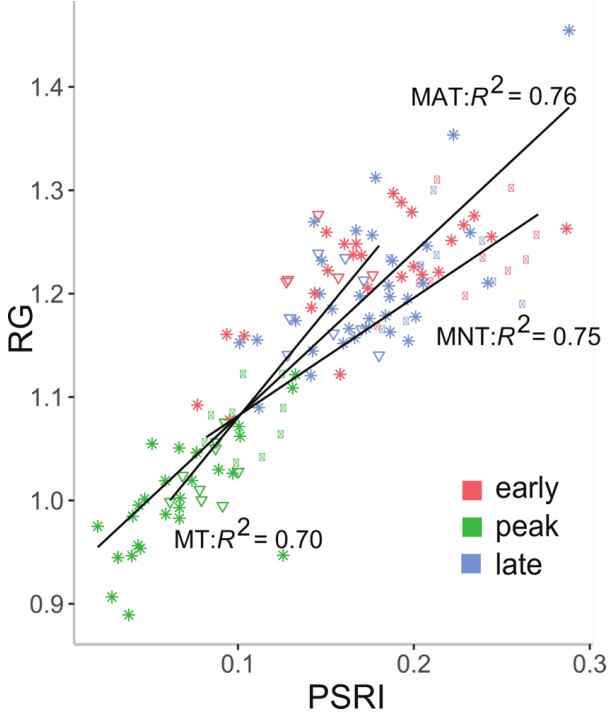

**Figure 17.** The relationships between pigment driven spectral indices of Plant Senescence Reflectance Index (PSRI) calculated from canopy level spectral reflectance and the simple normalized Red Green index calculated from the digital camera data in three dominant vegetation communities. MNT: Moist non-acidic tundra; MAT: Moist acidic tundra; MT: Moss tundra. PSRI is used to track plant senescence related to the degradation of chlorophyll pigments (Beamish et al., 2018).





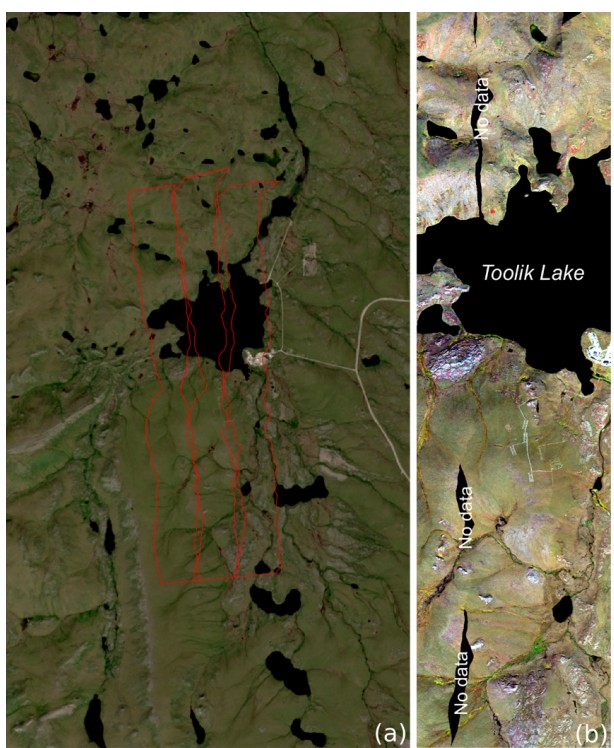


**Figure 18.** Toolik Lake, Alaska. (a) Sentinel-2 image (21.07.2016) with airborne hyperspectral flight lines in red; (b) AISA Eagle hyperspectral image (1.3 m) (Beamish et al., 2019b).





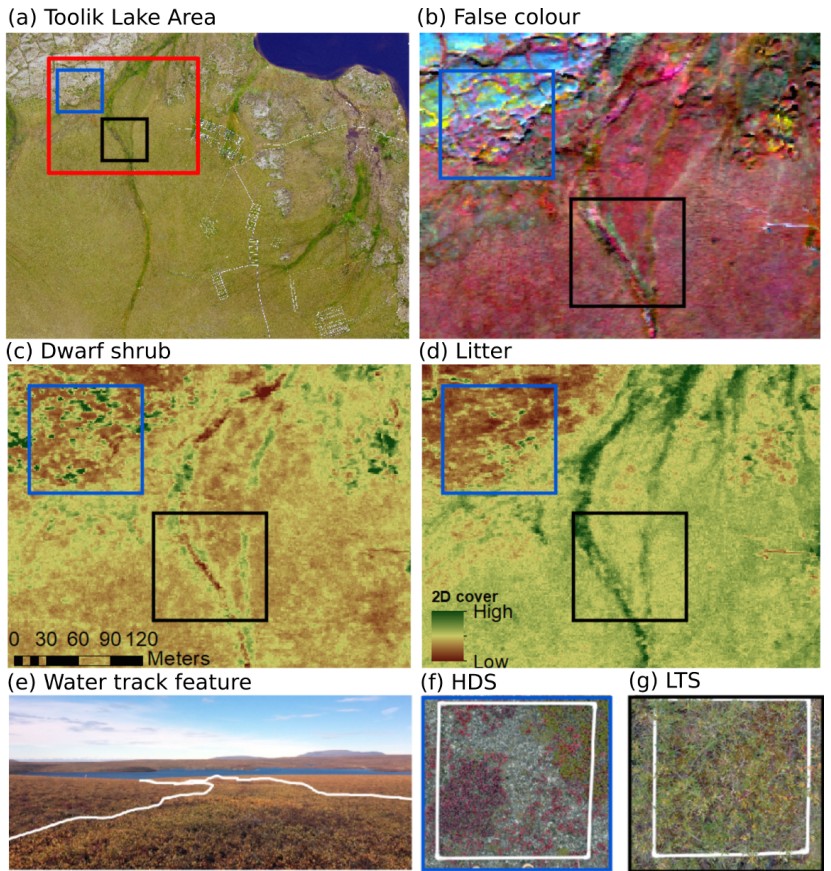

**Figure 19. a)** A 50 cm aerial photograph of the Toolik vegetation grid (Toolik Lake Environmental Data Centre, https://toolik.
alaska.edu/edc/, 2017), inset represents the extent of hyperspectral imagery below; b) a false color composite of the north west
corner of the study area, R: 812, G: 686, and B: 546 nm; c) Dwarf shrub 2D cover estimated by NRI λ1: 550 λ2: 650 nm; d)
Litter 2D cover estimated by NRI λ1: 525 and λ2: 600 nm; e) field photo of a water track feature from the study area; f) Hemi-
prostrate and prostrate dwarf-shrub, forb, moss, fruticose-lichen tundra; g) Low and tall shrublands (Beamish et al., 2019b).
The airborne image is provided by ©Toolik Lake Environmental Data Centre (https://toolik. alaska.edu/edc/, 2017).





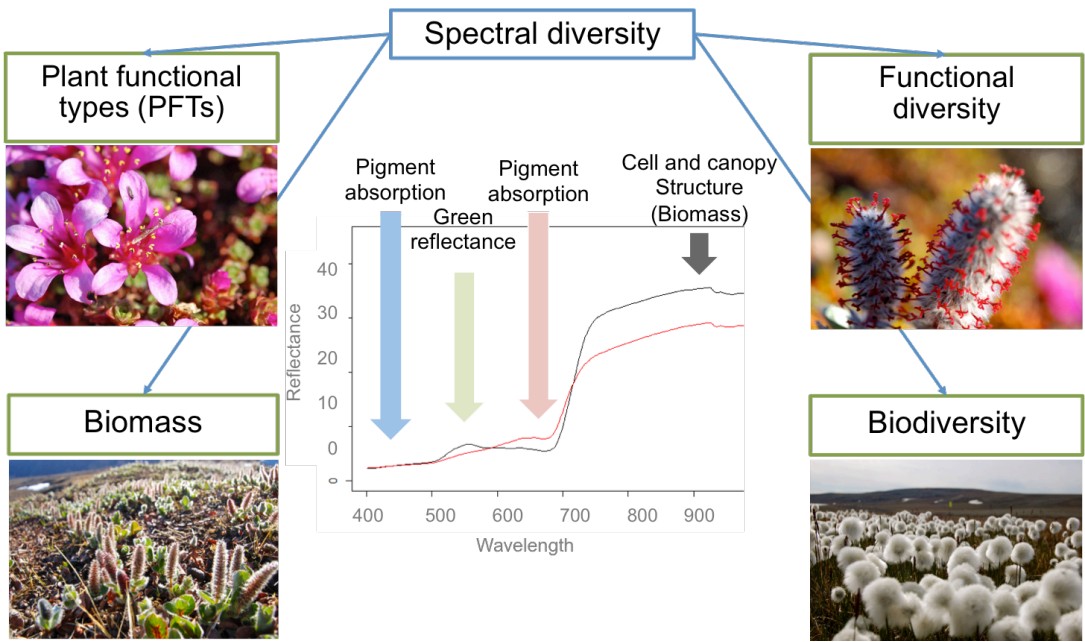

**Figure 20.** Spectral characterization of key biophysical and biochemical variables can inform interpretation of airborne and satellite hyperspectral remote sensing.

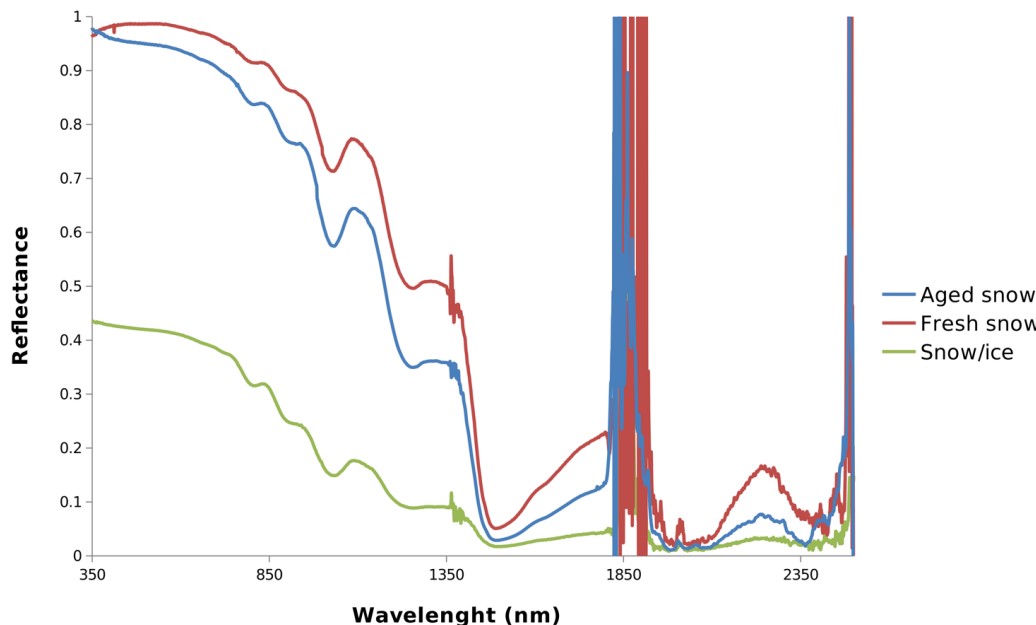

**Figure 21.** Spectral behaviour of different snow covers.



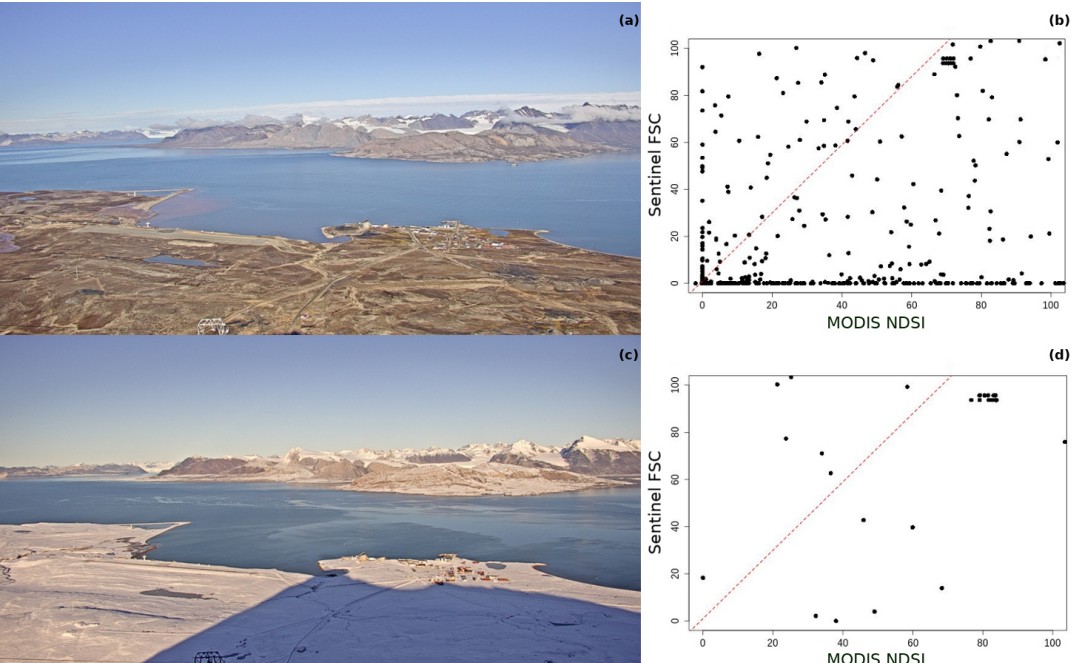

**Figure 22.** The snow cover at Ny-Ålesund during two different seasons: (a) August 20th 2018; (c) September 17th 2018. Relationship between the spectral index of snow (NDSI) obtained by MODIS data, and the Fractional Snow Cover estimated by Sentinel-2 and detected by terrestrial photography (b,d) (Pedersen, 2013).

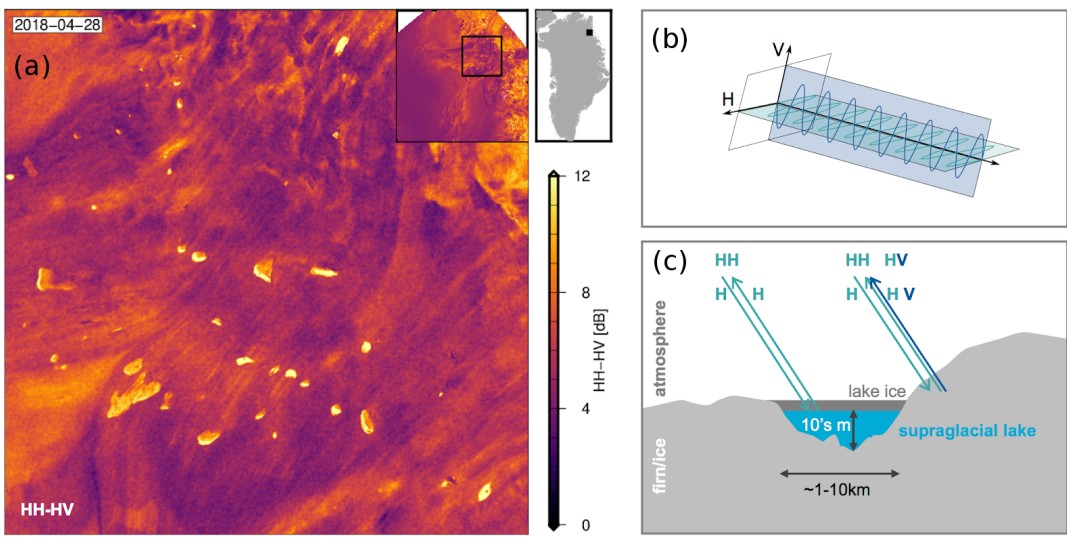


**Figure 23. (a)** Detection of supraglacial lakes using polarimetry using Sentinel-1. Bright yellow colors denote surface scattering, which is arising from the flat transition between lake ice and water, as shown schematically in the right panel. **(b)** and **(c)** a schematic representation of principle of polarimetric method.



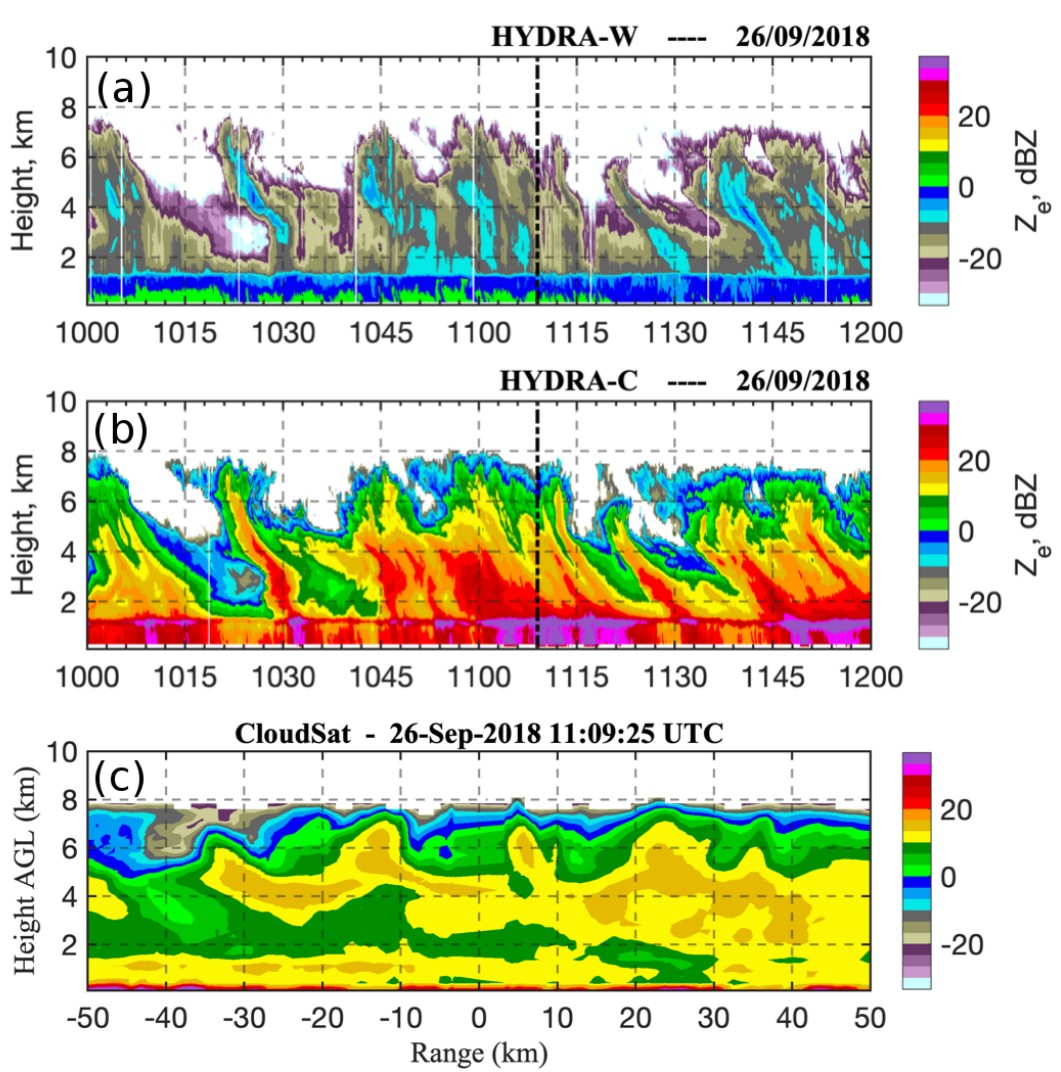

**Figure 24.** Observations of clouds and precipitation using Hyytiälä **(a)** cloud, **(b)** precipitation radars and **(c)** CloudSat cloud profiling radar, the bottom panel. The vertical dashed line in the panels indicates the CloudSat overpass time. The x axis in the CloudSat panel shows the distance between the Hyytiälä site and the satellite ground-track.

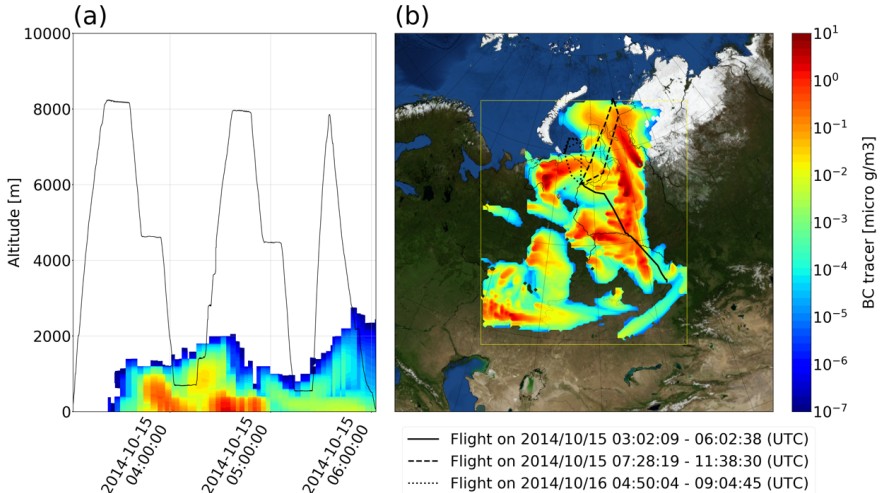

**Figure 25.** Simulated concentrations of BC tracer with 5 day lifetime in micro g m$^{-3}$ **(a)** along YAK flight on 15 October 2014 (flight altitude also given, white line), and **(b)** at the surface at 05 UTC on 15 October 2014, flight tracks are also shown. BC tracer concentrations are normalised to the total BC emissions from Huang et al. (2015) over the model domain and emitted based on annual mean flaring locations using VIIRS data.

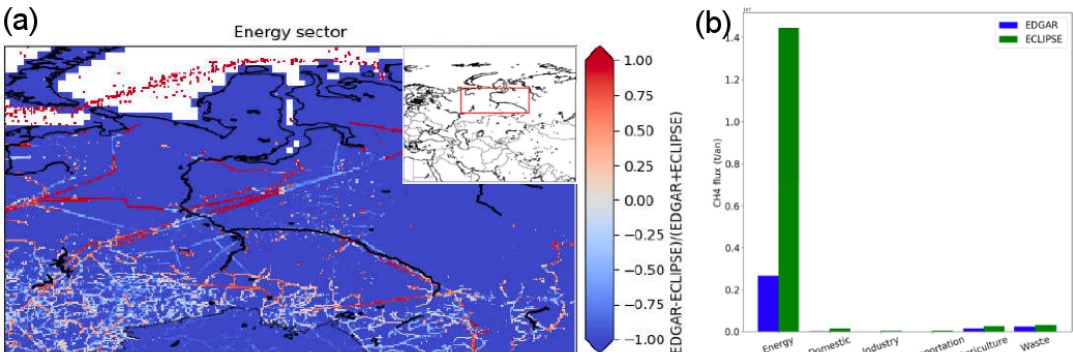

**Figure 26.** Comparison of the EDGAR v4.3.2 and ECLIPSE v5a inventories for methane. **(a)** spatial distribution normalized difference **(b)** total by sector of methane emission for the western Siberia area.



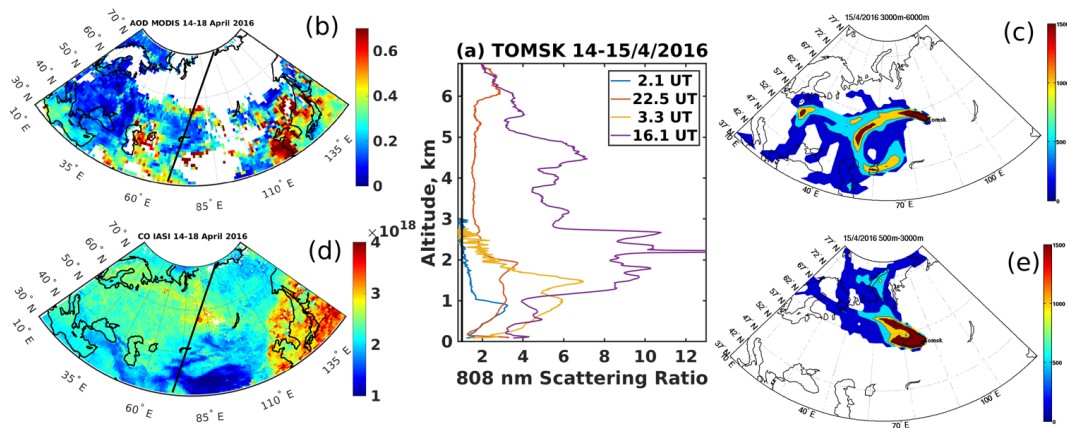

**Figure 27.** Vertical profiles of scattering ratio on 14 and 15 April 2016 from Tomsk lidar **(a)** and corresponding 5-day average map of AOD at 532 nm from MODIS **(b)** and of CO total column in molecules cm$^{-2}$ from IASI observations **(d)**. The red circle is Tomsk and the black thick lines are the CALIOP overpass over Tomsk on April 14 at 21 UTC. Potential emission sensitivity (PES) distributions for FLEXPART backward simulations initialized below 3 km **(e)** and above 3 km **(c)** are shown for 15 April, 2016.

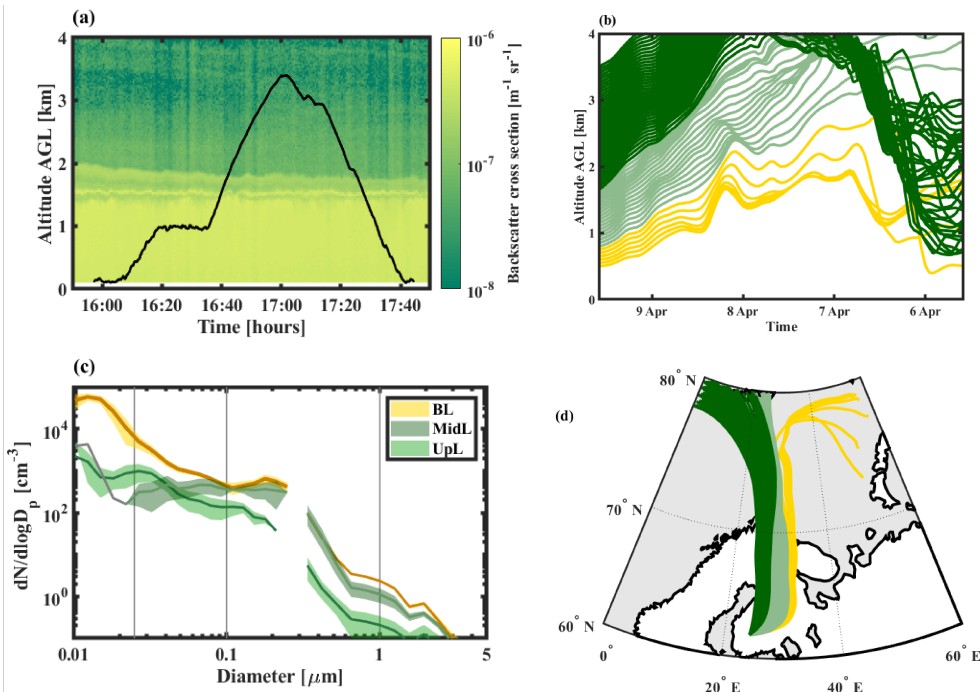

**Figure 28.** Clear sky case study during 9 April 2014 at SMEAR II station, Finland: **(a)** HSRL backscatter coefficient with Cessna flight altitudes in black; **(b)** 96h backward trajectories calculated every 50 m and combined into layers according to similarities in the travelling path; **(c)** aerosol size distribution measured onboard during the flight and combined into layers, shown with 1 standard deviation; **(d)** spatial coverage of backward trajectories.

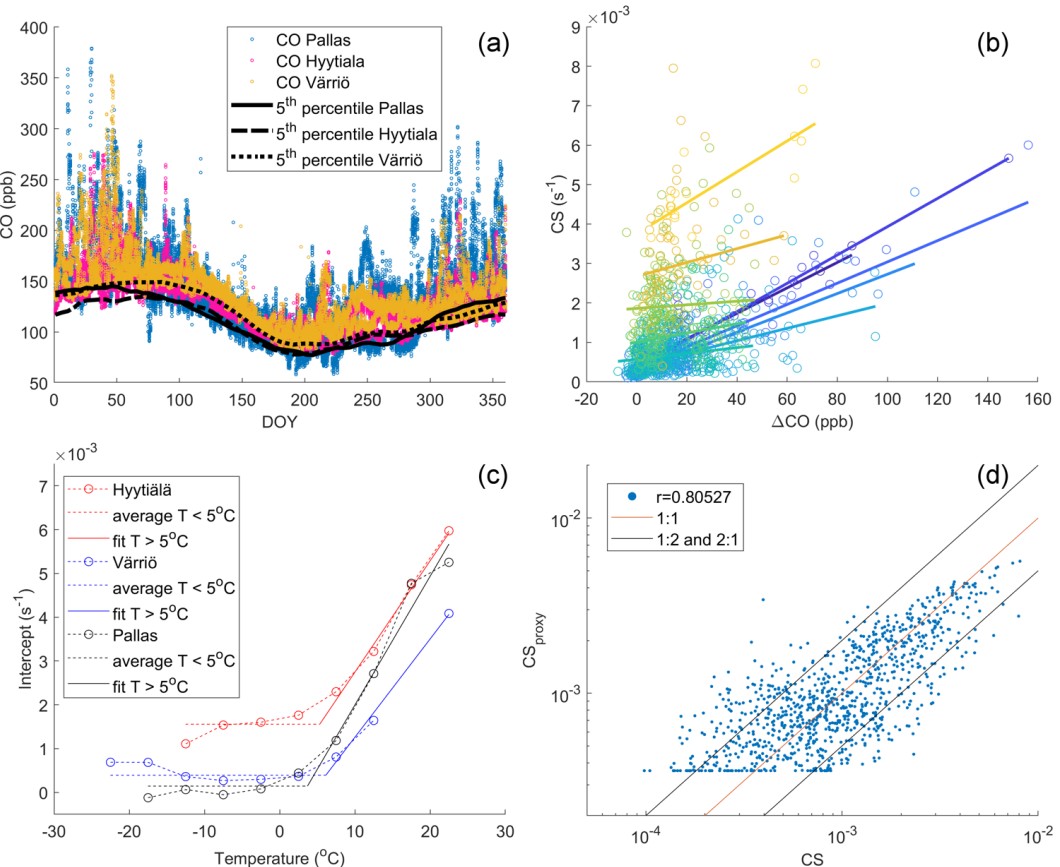

**Figure 29.** Derivation of proxy for condensation sink (CS) based on carbon monoxide (CO) concentration and air temperature.
**(a)** annual pattern of CO concentration and its 5th percentiles at the observation sites, **(b)** observed CS in Värriö as a function
of the difference between observed CO and the 5th percentiles from panel (a) in 5 °C temperature bins (blue colors for T < 5°C,
green and yellow for T > 5°C), **(c)** intercepts $a_i(T)$ in fittings to data in panel (b), when slopes are forced to their average $b_{ave}$,
as functions of temperature, and **(d)** the resulting proxy as shown in Eq. 3.


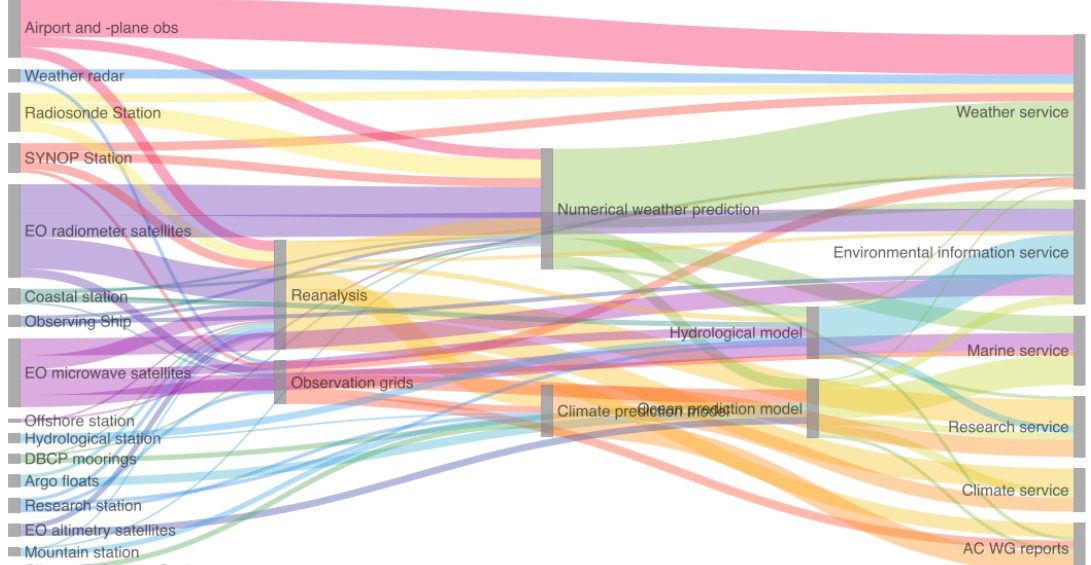

**Figure 30.** The observations, modelling and services parts of the value tree - the source tree. iCUPE actions are represented

1795 in research stations, EO satellites and observation grids supporting research services. The framework foresees 3 levels, but

modelling is feeding its own components, which adds more depth to the tree.



**Table 1.** iCUPE products as datasets resulted from research activities of the project as well as collaboration with PEEX Russian partners. First dataset was delivered in December 2018.

| N | iCUPE products as datasets on: |
|---|---|
| 1 | Emerging organic contaminants in air from the Arctic |
| 2 | Ground based measurements for particle number, black carbon mass and ozone concentration |
| 3 | Black carbon and aerosol absorption of Arctic research infrastructures |
| 4 | Anthropogenic contaminants in snow from polar regions |
| 5 | Anthropogenic contaminants in ice cores |
| 6 | Emerging organic contaminants in snow from the Arctic |
| 7 | Emerging organic contaminants in water from the Arctic |
| 8 | Snow spectral reflectance |
| 9 | Aerosol vertical profiles from ground-based and satellite observations in Finland and Russia |
| 10 | Arctic atmospheric mercury observations: updated GMOS database |
| 11 | Blueprint for novel proxy variables integrating in-situ and satellite data |
| 12 | Arctic parameters based on ground-based remote sensing and airborne platforms |
| 13 | Precipitation in the high-latitudes |
| 14 | Novel optical remote sensing products on snow and on vegetation and gas flaring mapping in selected sites |
| 15 | Arctic atmospheric mercury isotope observations |
| 16 | Time series of lakes' size changes in Northeast Greenland |
| 17 | Aerosol reanalysis for SMEAR-II |
| 18 | Organic aerosols in the Arctic based on source apportionment |
| | **Datasets resulted from collaboration with PEEX Russian partners** |
| 20 | Mercury measurements at Amderma station of the Russian Arctic |
| 21 | Elemental and organic carbon over the northwestern coast of the Kandalaksha Bay of the White Sea |
| 22 | Micro-climatic features and Urban Heat Island Intensity in cities of the Arctic region |
| 23 | Atmospheric composition at Fonovaya Observatory, West Siberia |

1800

**Table 2.** WMO Oscar statistics on operational stations in 30° latitude slices, iCUPE stations are mostly non-affiliated

| Latitude slice | total stations | WIGOS | co-sponsored | non-affiliated | area % | land % |
|---|---|---|---|---|---|---|
| 60°N-90°N | 2 218 | 1 352 | 853 | 13 | 6,53 | 3,04 |
| 30°N-60°N | 13 070 | 7 436 | 5 472 | 162 | 18,37 | 8,79 |
| 0°N-30°N | 8 062 | 2 820 | 5 222 | 20 | 25,1 | 6,81 |
| 0°S-30°S | 5 805 | 2 300 | 3 432 | 73 | 25,1 | 5,66 |
| 30°S-60°S | 4 120 | 917 | 3 199 | 4 | 18,37 | 0,91 |
| 60°S-90°S | 712 | 165 | 547 | 0 | 6,53 | 2,53 |