# Peer review of "Integrative and comprehensive Understanding on Polar Environments (iCUPE): the concept and initial results"

_Atmospheric Chemistry and Physics, 2019_

## Referee Comment (RC1) · Anonymous Referee #1 · 20 Mar 2020

Review of the manuscript, "Integrative and comprehensive Understanding on Polar Environments (iCUPE): the concept and initial results," by T Petaja et al., submitted to Atmosphere Chemistry and Physics.

(General comments) At first, I felt some difficulties to understand the manuscript as "Research article" which should "report substantial new results and conclusions from scientific investigation,..." as expressed in the Manuscript types written on ACP Home Page. The manuscript is rather "Review" or "Overview article" for the special issue. Actually, it is indicated as the articles for the special issue, "Pan-Eurasian Experiment (PEEX)", and the manuscript most fits as "Special issue overview article".

[Figure]

Even though, I was confused to the substance of the project "iCUPE", if it conducts observation itself or just works for analyzing activities, which are not clearly mentioned in the manuscript. I have found some expression in the iCUPE home page; iCUPE will 1) synthesize data from comprehensive long-term measurements, intensive campaigns and satellites, collected during the project or provided by on-going international initiatives, which clearly mentions the actual activities of the project. Please add this kind of explanation in the manuscript, then, it will be much understandable.

The manuscript is not well organized, all of the substances are written in chapter 3, and still the sections are mixtures of methods (3.1, 3.2, 3.3, 3.7 and 3.8) and target species (3.4, 3.5 and 3.6), so, not easy to read and understand, partly, also, due to the question in the previous paragraph. 3.8.1 is also very difficult to follow, since the substances are cloud (microphysics) and precipitation, which are quite far from other items discussed in the manuscript.

(Specific coments) - For figures which were not the original of this paper, the citation should be shown.

- Papers which were not published yet should not cited, such as "to be submitted", "submitted", "in preparation", and so on. I am not sure for the paper "in press."

- Line 218-220: The sentence "When the Polar Front retreats, anthropogenic emissions are no longer able to penetrate into the High Arctic" is miss leading. -→ ..., even anthropogenic emissions penetrate into the High Arctic, they could not kept as high concentration.

- 3.2.2. Black Carbon: What is the equivalent black carbon concentration (eBC)? There is no explanation here. We could not access to the paper by Kalogridis (2019), which is just "to be submitted". There is no explanation for the correction to the aethalometer BC concentration proposed by Sinha et al. (2017, JGR). It was reported that BC concentrations measured by aethelometer (Sharma et al., 2013) or by PSAP (Hirdman et al., 2010) at Ny-Alesund were 20 – 30 % larger compared to the value obtained by

COSMOS (back upped by SP2). This is also in Fig. 5.

- Line 729-733: Validation of satellite cloud profiling radar by comparing with the ground based radar, as in Fig. 24 is not clear. It is better to compare the vertical profile from both the radars.

- Are the greenhouse gases not the targets of the project? Only atmospheric trace gases are expressed in line 865-866. Methane anthropogenic emission is discussed in 3.8.2.
* * *

---

## Referee Comment (RC2) · Anonymous Referee #2 · 28 Apr 2020

The paper of Petäjä et al. summarizes the aims and preliminary results of a project called "iCUPE - integrative and Comprehensive Understanding on Polar Environments". The main objective of the project is to provide assessment of observational data (ground based, remote, and satellite) with focus on Arctic. There is no doubt that the targeted issue is very relevant and the applied methodologies are adequate. Because of the different aspects of the significance in the research field, the subject of the paper is of great interest not only to the atmospheric and aerosol physics community, but also for a boarder audience. Nevertheless, after reading the manuscript I am not convinced that the correct manuscript type of the present paper is Research article. Although there are a lot of novel results and scientific conclusions listed, the detailed

scientific deduction of single methods, results, discussions, and conclusions are missing. Therefore, in my opinion the manuscript is rather an Overview or a Review article. However, for an Overview article the substantial results are not listed and discussed in the adequate way, and only a few new papers and submissions from the project are cited (the most in Sections 3.7.3; 3.7.2). It is also not clear for the reader what iCUPE exactly is. When has it started? When will it end? For me the manuscript sounds rather as a progress report than a scientific research paper. This is also reflected in the non-uniform presentation of figures and structure of sections. I suggest that the Authors define at the end of the Introduction what they will present in the paper and why those issues are important. In the current stage, topics are just following each other. The only keyword connecting the topics is "Arctic", I could not recognize any other links among the sub-areas. As I mentioned, the paper is nonuniformly structured and written, there are some section which are very well written (Section 3.3.) or presented in a very detailed manner (e.g., Section 3.7.2.). The Authors should pay heed that all figures are understood without reading the text, and that they are described adequately in the text. . This is definitely not the case for Figs. 1, 2, and 3. Furthermore, most of the figures are discussed poorly in the text. The most detailed discussion is provided for Fig. 30; including costs – is that really an issue for a scientific paper?

In the following I list some minor comments to be considered when revising the manuscript:

Reference to Figs. 21, and 25 are missing in the paper

Page 7, line 248: What are the ACTRIS and IASOA networks? And how are they connected to Figure 3?

Page 7, discussion about Fig. 5: the gradual reduction mentioned is not obvious. Probably a trend line would help.

Page 8, line 270: Where is the simulation presented? And how? This is not clear here.

Page 8, line 280: Complex sentence, hard to understand, and to follow. The statements here are not supported by concrete data or figure. Please refer them here.

Line 355/368: Aladina was flying up to which level? 800 or 850 m?

What is the red area in Fig 9.?

Figure 11b: In the figure different metrics are used for fluxes and reservoirs, so I suggest different style/color for the numbers.

Page 13, line 470: the linear approximation was carried out on Fig. 14a, and not on Fig. 14b, when I understand correctly.

Lines 479-484: I just highlight this part as a prominent example making the reader confused. Have you done that already, or do you plan to do that in the future, or you did it but published elsewhere, or is this just for posing the problem?

Page 19, line 731: Fig. 24: Dash-dotted line indicates the CloudSat overpass, not a dashed line which is used for the grid lines. Further, the difference in the observed values is not obvious for a non-expert reader.

Page 19, line: 737: Again, just as a prominent example. Does iCUPE only models/analyses/uses the data collected from 2014 till 2017, or was that the part of the project?

Fig. 27: the details given for the model in the text should also been provided to figure caption.

Page 20, line 805: I think the lowest size limit for the aerosol measurement is 0.01 micron. At least this is what I suppose from Fig. 28.

Page 23: Some part of the data flow should be written somewhere else in a statement at the end of the paper, not in the main part. For example, which platforms are planned to be tested.

In summary, I recommend the publication of the manuscript in ACP after major revision. I suggest the authors to read the complete manuscript carefully and try to understand the scientific statements of the co-authors. Please feel free to criticize and ask questions. I understand that this is a huge amount of work, but that would definitely improve the quality of the paper and help non-expert readers to understand the presented results of this very important topic.
* * *

---

## Author Comment (AC1) · 20 May 2020

Author responses to Referee 1 comments on the paper "Integrative and comprehensive Understanding on Polar Environments (iCUPE): the concept and initial results" by Tuukka Petäjä et al.

The authors are grateful for the referee for the comments, which improved the manuscript considerably. We provide point-by-point responses below in **bold.**

(General comments) At first, I felt some difficulties to understand the manuscript as "Research article" which should "report substantial new results and conclusions from scientific investigation…" as expressed in the Manuscript types written on ACP Home Page. The manuscript is rather "Review" or "Overview article" for the special issue. Actually, it is indicated as the articles for the special issue, "Pan-Eurasian Experiment (PEEX)", and the manuscript most fits as "Special issue overview article".

**We agree with the referee. The more suitable type of the manuscript is "Special issue overview article". Upon submission, we were not able to connect the paper to the PEEX special issue and this was done only at the technical edit phase. We hope that the type can be changed in the next editorial phase.**

Even though, I was confused to the substance of the project "iCUPE", if it conducts observation itself or just works for analyzing activities, which are not clearly mentioned in the manuscript. I have found some expression in the iCUPE home page; iCUPE will 1) synthesize data from comprehensive long-term measurements, intensive campaigns and satellites, collected during the project or provided by on-going international initiatives, which clearly mentions the actual activities of the project. Please add this kind of explanation in the manuscript, then, it will be much understandable.

**The connection between the paper and the iCUPE project is now clarified in the last paragraph of the introduction section as follows (new text indicated with red):**

**"The iCUPE project aims to synthesize data from comprehensive long-term measurements, intensive campaigns and satellites, collected during the project or provided by on-going international initiatives.** **The aim of this paper is to introduce an on-going project iCUPE and summarize its initial results. We put a specific emphasis on black carbon and persistent pollutants in the Arctic context. We explore snow and ice core samples to put the current concentrations in longer perspective. We underline the capacity of the continuous observations to monitor the impact of policies to reduce the emissions. We showcase the potential to address the pollution in the Arctic environment by integrating satellite remote sensing, airborne observations, in situ data and modeling. The modern comprehensive source apportionment can resolve the different sources of atmospheric aerosols and differentiate between sources within and outside the Arctic environment. We also discuss the iCUPE impact and relevance for the Arctic research and for the stakeholder communities."**

The manuscript is not well organized, all of the substances are written in chapter 3, and still the sections are mixtures of methods (3.1, 3.2, 3.3, 3.7 and 3.8) and target species (3.4, 3.5 and 3.6), so, not easy to read and understand, partly, also, due to the question in the previous paragraph. 3.8.1 is also very difficult to follow, since the substances are cloud (microphysics) and precipitation, which are quite far from other items discussed in the manuscript.

**It is true that the manuscript structure is not optimal. However, this reflects the variety of topics addressed by the iCUPE consortium. We organized section 3 with the following logic:**

We start with the in-situ component of iCUPE with sections 3.1-3.6 and bring in to discussion the satellite remote sensing in section 3.7. Then we present selected integrating examples in 3.8, which include also method development and conceptualization. This is consistent with the iCUPE concept as a whole (multiplatform observations, modeling, synthesis, summarized in Figure 2).

We include a short explanatory paragraph in Section 3 as follows:

"In this section we summarize results and findings of iCUPE-project regarding the in-situ observations (Sect 3.1-3.6) and the satellite component (Sect. 3.7). Then we selected integrating examples in Sect. 3.8, which include also method development and conceptualization. This is consistent with the iCUPE concept as a whole (multiplatform observations, modeling, synthesis, Figure 2)."

We edited the subsection names for 3.2 and 3.3 to be consistent with the others (removed word "Results").

(Specific comments)

For figures which were not the original of this paper, the citation should be shown.

**We have included the references to the original work, when applicable.**

- Papers which were not published yet should not cited, such as "to be submitted", "submitted", "in preparation", and so on. I am not sure for the paper "in press."

**We have updated the reference list and only published papers are listed.**

- Line 218-220: The sentence "When the Polar Front retreats, anthropogenic emissions are no longer able to penetrate into the High Arctic" is missleading. -→ ..., even anthropogenic emissions penetrate into the High Arctic, they could not kept as high concentration.

**We clarified and edited the paragraph as follows:**

"When the Polar Front retreats, transport of anthropogenic emissions is limited to the High Arctic as only emissions north of the Polar Front can find a direct way to high Arctic sites and the front is located much further north. Also, wet removal processes reduce the build up of high concentrations. Other transport processes would only be pathways in high altitudes and then penetrating the Polar Dome by descending air masses through entrainment reaching high Arctic sites, which is rarely being observed."

- 3.2.2. Black Carbon: What is the equivalent black carbon concentration (eBC)? There is no explanation here. We could not access to the paper by Kalogridis (2019), which is just "to be submitted". There is no explanation for the correction to the aethalometer BC concentration proposed by Sinha et al. (2017, JGR). It was reported that BC concentrations measured by aethelometer (Sharma et al., 2013) or by PSAP (Hirdman et al., 2010) at Ny-Alesund were 20 – 30 % larger compared to the value obtained by C2. This is also in Fig. 5.

The definition of equivalent black Carbon concentration is given in the text with the appropriate reference. The reference to Kalogridis (2019) was removed as the submission is still pending.

The eBC data here are consistent with all previous publications where the specific mass absorption efficiency of the instrument is used. When the data are reported as aerosol absorption coefficients, then the compensation algorithms proposed in the literature can be used. The different measurement techniques undergo separate treatment and corrections which are specific to the various instruments. Here we do not employ any data obtained by the Ny Aalesund PSAP and it is not appropriate to refer to findings regarding this instrument. We do have evidence from a short study on the relationship among several instruments at then Zeppelin station, where the variability in the multiple scattering compensation parameter C is discussed but as mentioned above this is applied when the data are reported in terms of the aerosol absorption coefficient. The loading effect compensation for the Aethalometer at the Zeppelin station was found in Zanatta et al, 2018 to be an insignificant source of uncertainty. This reference and the work by Backman et al., 2017 contains all the information regarding the currently available quality assurance for the aethalometers at Ny Ålesund and is added instead of (Kalogridis, 2019).

As a result, the two first paragraphs in the section 3.2.2. 3.2.2 Black Carbon concentrations at Mt Zeppelin, Svalbard, was edited as follows:

"Black Carbon (BC) is one of the key short-lived climate forcers contributing to the warming of the Arctic both by absorbing the solar radiation but also by enhancing snow and ice melt by surface deposition (e.g. Bond et al., 2013). As part of the atmospheric observations, the ACTRIS and IASOA networks (Fig. 3) operates a network of aethalometers to determine the atmospheric concentration of BC in the air (Uttal et al., 2016). Although BC is the common term we use for light absorbing carbon, it is now more appropriate to report mass concentrations in terms of equivalent black carbon (eBC), especially when filter-based optical techniques are employed. Equivalent black carbon mass concentration is considered to be the mass of an equivalent amount of light absorbing carbon with a given mass absorption efficiency causing the attenuation of light observed by the instrument at a given wavelength (Petzold, 2013). The quality assurance of data for eBC and the corresponding aerosol absorption coefficient has greatly improved over the last years. Compensation schemes for measurement artifacts as well as harmonization of data obtained by different instruments have been established (Backman et al., 2017) and are continuously updated (Zanatta et al., 2018).

The results from long-term observations at Zeppelin have been discussed and assessed in several works elaborating on the climatology of BC in the Arctic i.e. (Eleftheriadis et al., 2009.; Sharma et al., 2013; Breider et al., 2017; Schmeisser et al., 2018). The results presented here is the longest continuous eBC reported record by a single instrument in the European High Arctic (Torseth at al., 2019) and globally the second longest after those obtained between 1989-2009 in Alert (Sharma et al., 2019). As an example of long-term observations of BC, we show the latest eBC concentration time series from Zeppelin Station at Svalbard (Fig. 5). The results show a continued gradual reduction in the annual mean value of observed eBC, while the time series is strongly modulated by a seasonal cycle well known in the Arctic with minima in the summer and maximum in late winter spring. One can observe this long-term decline with a linear trend line applied only as a crude estimate for these data. The long-term data series we present here makes it possible to derive some descriptive statistics. The eBC annual mean value has been reduced from an annual mean value of31 ng m$^{-3}$ at the beginning of the previous decade to 12

ng m$^{-3}$ during the last years with an average reduction of 7 ng m$^{-3}$ per decade which amounts to a reduction 4% annually or approximately 44% per decade. However, trend analysis for aerosol climatology records need to be practiced with caution in order to remove the effects of the seasonal cycle. When the extracted absorption coefficient from our data was thoroughly examined for a shorter period (2005-2018) the trend was not found to be statistically significant (Collaud Coen et al., 2020). Minimum values over the summer often drop below the detection limits of the instrument while maximum values vary greatly with their occurrence usually related to large scale biomass burning events across Siberia and Alaska. The continuous reduction in fossil fuel usage is a reason for this reduction but it is well known that emissions are not uniformly changing on a global scale or at least in the Northern hemisphere (Evangeliou et al., 2018)."

- Line 729-733: Validation of satellite cloud profiling radar by comparing with the ground based radar, as in Fig. 24 is not clear. It is better to compare the vertical profile from both the radars.

**We have updated the text to made a clearer explanation of the Fig. 24 as follows:**

"To prepare for the upcoming ESA Earthcare validation activities, preparatory studies were performed on how ground-based radar observations can be used to validate space-based radar observations of vertical profiles of clouds and precipitation. A comparison between such observations, in this case we use CloudSat, is shown in Fig. 24. The figure shows vertical profiles of radar reflectivity as observed using ground-based and satellite-based cloud radars. As can be noticed there are detectable differences in the observed values. The CloudSat observed values are higher at cloud tops and lower in precipitation if compared to the ground-based radar. These differences are caused by attenuation of ground-based radar measurements in rain and melting layer and in cloud and melting layer for CloudSat observations. Therefore, a direct comparison of observed vertical profiles requires a method that can into account the attenuation. As a part of iCUPE Li et al., (2019) has studied the impact of melting layer of precipitation on cloud radar observations. The results of this study will be used for the EarthCare validation in the future."

- Are the greenhouse gases not the targets of the project? Only atmospheric trace gases are expressed in line 865-866. Methane anthropogenic emission is discussed in 3.8.2.

**The emissions of greenhouse gas emissions are not a specific target of the project. In our terminology, we prefer to use "trace gases", which include also the greenhouse gases, such as $CO_2$ and methane as they are in the atmosphere in trace amounts.**

**References**

Petzold, A., Ogren, J. A., Fiebig, M., Laj, P., Li, S.-M., Baltensperger, U., Holzer-Popp, T., Kinne, S., Pappalardo, G., Sugimoto, N., Wehrli, C., Wiedensohler, A., and Zhang, X.-Y.: Recommendations for reporting "black carbon" measurements, Atmos. Chem. Phys., 13, 8365–8379, https://doi.org/10.5194/acp-13-8365-2013, 2013.

Zanatta, M., Laj, P., Gysel, M., Baltensperger, U., Vratolis, S., Eleftheriadis, K., Kondo, Y., Dubuisson, P., Winiarek, V., Kazadzis, S., Tunved, P., and Jacobi, H.-W.: Effects of mixing state on optical and radiative properties of black carbon in the European Arctic, Atmos. Chem. Phys., 18, 14037–14057, https://doi.org/10.5194/acp-18-14037-2018, 2018.

---

## Author Comment (AC2) · 20 May 2020

Author responses to Referee 2 comments on the paper "Integrative and comprehensive Understanding on Polar Environments (iCUPE): the concept and initial results" by Tuukka Petäjä et al.

The authors are grateful for the referee for the comments, which improved the manuscript considerably. We provide point-by-point responses below in **bold** and the changes in the manuscript are in **red.**

The paper of Petäjä et al. summarizes the aims and preliminary results of a project called "iCUPE - integrative and Comprehensive Understanding on Polar Environments". The main objective of the project is to provide assessment of observational data (ground based, remote, and satellite) with focus on Arctic. There is no doubt that the targeted issue is very relevant and the applied methodologies are adequate. Because of the different aspects of the significance in the research field, the subject of the paper is of great interest not only to the atmospheric and aerosol physics community, but also for a boarder audience. Nevertheless, after reading the manuscript I am not convinced that the correct manuscript type of the present paper is Research article. Although there are a lot of novel results and scientific conclusions listed, the detailed scientific deduction of single methods, results, discussions, and conclusions are missing. Therefore, in my opinion the manuscript is rather an Overview or a Review article.

**We agree with the referee. As already pointed out in our response to the referee 1, The more suitable type of the manuscript is "Special issue overview article". Upon submission, we were not able to connect the paper to the PEEX special issue and this was done only at the technical edit phase. We hope that the type can be changed in the next editorial phase.**

However, for an Overview article the substantial results are not listed and discussed in the adequate way, and only a few new papers and submissions from the project are cited (the most in Sections 3.7.3; 3.7.2). It is also not clear for the reader what iCUPE exactly is. When has it started? When will it end? For me the manuscript sounds rather as a progress report than a scientific research paper. This is also reflected in the nonuniform presentation of figures and structure of sections. I suggest that the Authors define at the end of the Introduction what they will present in the paper and why those issues are important.

**We have clarified iCUPE in the last paragraph of the introduction with the following edits:**

**"The iCUPE project aims to synthesize data from comprehensive long-term measurements, intensive campaigns and satellites, collected during the project or provided by on-going international initiatives."**

In the current stage, topics are just following each other. The only keyword connecting the topics is "Arctic", I could not recognize any other links among the sub-areas. As I mentioned, the paper is nonuniformly structured and written, there are some section which are very well written (Section 3.3.) or presented in a very detailed manner (e.g., Section 3.7.2.).

**We included a short paragraph at the beginning of the results section to justify the order of the sections and to help the reader to be able to follow our line of thoughts:**

**"In this section we summarize results and findings of iCUPE-project regarding the in-situ observations (Sect 3.1-3.6) and the satellite component (Sect 3.7). Then we selected integrating**

**examples in Sect. 3.8, which include also method development and conceptualization. This is consistent with the iCUPE concept as a whole (multiplatform observations, modeling, synthesis, Figure 2)."**

The Authors should pay heed that all figures are understood without reading the text, and that they are described adequately in the text. This is definitely not the case for Figs. 1, 2, and 3.

**We have edited the figure captions for clarity as follows:**

**Figure 1. A conceptual figure on processes affecting the atmospheric composition in the Arctic. Atmospheric concentration of pollutants and their lifecycle in high latitudes are affected by local and regional anthropogenic activities and long-range transport from lower latitudes. Pollutant distributions and life cycles are modulated by transport patterns, changes in the biosphere, increased natural resource extraction and increased shipping in the Arctic Sea. Various feedbacks and interactions can either speed up or hinder the changes.**

**Figure 2. Integration of in-situ observations, satellite remote sensing and multi-scale modeling is required to provide a broad view on the current status of the environment as a whole. The multi-platform observational data and model data are cross-validated against each other. The integrative concept of iCUPE incorporates data and knowledge from ground-based observations, satellite remote sensing and modelling results providing a comprehensive view about the state of the environment in the polar areas.**

**Figure 3. A map of atmospheric aerosol observation stations with year-round observations within the Arctic provide a circumpolar view on the aerosol concentrations in the Arctic domain.**

Furthermore, most of the figures are discussed poorly in the text.

**We have clarified the figures and their connections to the text throughout the manuscript.**

The most detailed discussion is provided for Fig. 30; including costs – is that really an issue for a scientific paper?

**It is true that the value of the observation network rarely comes up in the discussion as a part of a scientific paper. However, the cost-benefit analysis is a valuable tool to address, on one hand, the costs of the Arctic observation system and, on the other hand, the societal benefits arising from these activities. This is depicted in Fig. 30 of the current paper, representing a value tree diagram and the width of connections represents the costs of the nodes in the value chain. Within the iCUPE project we performed analysis on the societal benefit of Arctic observations. The work is in progress, but it will lead to a thorough cost-benefit analysis. Our work underlines the capacity to perform a yearly cost analysis for the thousands of different stations and many satellites as well as modelling system involved to produce key information for societal benefit areas.**

In the following I list some minor comments to be considered when revising the manuscript:

Reference to Figs. 21, and 25 are missing in the paper.

**The style to refer to the figures was unified. We added the missing references to Fig 6, Fig 21 and Fig 25.**

Page 7, line 248: What are the ACTRIS and IASOA networks? And how are they connected to Figure 3?

**The Figure 3 summarizes the observational capacity in the Arctic with respect to in-situ aerosol observations. The observational networks contribute to many networks, such as ACTRIS, WMO-GAW, SMEAR and IASOA networks. In the sake of clarity, the specific contributions to the particular networks are not shown in Figure 3 as they tend to change over time.**

Page 7, discussion about Fig. 5: the gradual reduction mentioned is not obvious. Probably a trend line would help.

**We have rewritten the paragraph with some extra information and added some additional evidence from a recent dedicated study on trends with part of our data. As described in the text an assumed linear trend although not acceptable as a rigorous trend analysis provides the reader the evidence of the reduction. Despite the fact that this is not visually obvious (now more evident with the help of the dashed line) an estimate is also given as a percentage per decade in the text.**

Page 8, line 270: Where is the simulation presented? And how? This is not clear here.

**We clarified the sentence and added a reference to Fig 6:**

**The analysis is done using reanalysis meteorological inputs from the European Center for Medium-range Weather Forecasts (ECMWF) on a resolution of one degree. The Potential Source Contribution Function (PSCF) is applied on both tracers. Western Siberia appears as the main source region on the PSCF analysis (Fig 6).**

Page 8, line 280: Complex sentence, hard to understand, and to follow. The statements here are not supported by concrete data or figure. Please refer them here.

**Clarified, see above.**

Line 355/368: Aladina was flying up to which level? 800 or 850 m? What is the red area in Fig 9.?

**For clarification, we removed the sentence starting as "ALADINA was operated from ground to 800 m a.g.l…."**

**The red area is now explained in the Fig. 9 caption:**

**"Flight path of the UAS ALADINA (start point in the center, marked in yellow) in order to study the vertical and horizontal distribution of aerosol particles above ice surfaces near glacier and over open water. Research flights were spatially limited by restricted areas (hatched in red), like bird sanctuary near coast, instrument areas south of Ny-Ålesund and no activity was performed**

**over inhabited buildings of the village. The TopoSvalbard map is obtained from Norwegian Polar Institute, retrieved from http://toposvalbard.npolar.no. ©"**

Figure 11b: In the figure different metrics are used for fluxes and reservoirs, so I suggest different style/color for the numbers.

**We edited the figure and clarified the difference with colors. The net fluxes (metric tons per year, in black) and the reservoirs (metric tons, in red) are now more easily differentiated in the figure.**

Page 13, line 470: the linear approximation was carried out on Fig. 14a, and not on Fig. 14b, when I understand correctly.

**The Fig 14a depicts the trend in duration of MADE whereas the Fig 14b describes the log-normal density function of the observations during different years.**

**We have clarified this in the text as follows:**

**"The histograms of the Hg0 concentration at Amderma during the different years is presented in Fig. 14b. The probability density distribution of the $Hg^0$ concentration was lognormal for the monitoring period from June 2010 to October 2013. There is a significant asymmetry in the left-hand region of the $Hg^0$ concentration probability distribution relative to the arithmetic average, pointing to the fact that low $Hg^0$ concentrations are measured more frequently. In 2013 this asymmetry was especially evident. The shift of the concentration to lower values was due to the increased amount of $Hg^0$ depletion events recorded during the winter seasons of 2010–2013. To assess the dynamics of $Hg^0$, a linear approximation of the average annual $Hg^0$ concentrations for the lognormal distribution was calculated with the reliability coefficient $R^2 = 0.7$ (Fig. 14b)**

**Furthermore, we removed the yellow background from the Figure 14b.**

Lines 479-484: I just highlight this part as a prominent example making the reader confused. Have you done that already, or do you plan to do that in the future, or you did it but published elsewhere, or is this just for posing the problem?

**Please see the comment above.**

Page 19, line 731: Fig. 24: Dash-dotted line indicates the CloudSat overpass, not a dashed line which is used for the grid lines. Further, the difference in the observed values is not obvious for a non-expert reader.

**The reference to the dash-dotted line was corrected. More explanation of the differences was also added as follows:**

 **"To prepare for the upcoming ESA Earthcare validation activities, preparatory studies were performed on how ground-based radar observations can be used to validate space-based radar observations of vertical profiles of clouds and precipitation. A comparison between such observations, in this case we use CloudSat, is shown in Fig. 24. The figure shows vertical profiles of radar reflectivity as observed using ground-based and satellite-based cloud radars. As can be noticed there are detectable differences in the observed values. The CloudSat**

**observed values are higher at cloud tops and lower in precipitation if compared to the ground-based radar. These differences are caused by attenuation of ground-based radar measurements in rain and melting layer and in cloud and melting layer for CloudSat observations. Therefore, a direct comparison of observed vertical profiles requires a method that can into account the attenuation. As a part of iCUPE Li et al., (2019) has studied the impact of melting layer of precipitation on cloud radar observations. The results of this study will be used for the EarthCare validation in the future."**

Page 19, line: 737: Again, just as a prominent example. Does iCUPE only models/analyses/uses the data collected from 2014 till 2017, or was that the part of the project?

**To clarify this, the last sentence of the paragraph was edited as follows:**

**Using these studies showed that the measurements collected at Hyytiälä, namely combined observations of the surface snowfall observations and multi-frequency radar observations, show the potential for verification of satellite cloud and precipitation retrieval algorithms in the future.**

Fig. 27: the details given for the model in the text should also been provided to figure caption.

**We edited the Figure 27 caption as follows:**

**"Figure 27. Vertical profiles of scattering ratio on 14 and 15 April 2016 from Tomsk lidar (a) and corresponding 5-day average map of AOD at 532 nm from MODIS (b) and of CO total column in molecules cm$^{-2}$ from IASI observations (d). The red circle is Tomsk and the black thick lines are the CALIOP overpass over Tomsk on April 14 at 21 UTC. Potential emission sensitivity (PES) distributions of an aerosol tracer averaged for 9 days before the Tomsk lidar aerosol layer observations below 3 km (e) and above 3 km (c) are calculated with backward simulations of the FLEXPART model using dry and wet deposition of the tracer during backward transport. "**

Page 20, line 805: I think the lowest size limit for the aerosol measurement is 0.01 micron. At least this is what I suppose from Fig. 28.

**Correct. This is described in line 815 as follows:**
**"A Scanning Mobility Particle Sizer (SMPS) and Optical Particle Sizer (OPS) were installed onboard Cessna FR172F aircraft (Schobesberger et al., 2013, Leino et al. 2019) to measure aerosol size distribution from 0.01 to 0.23 μm and 0.3 to 5 μm, respectively."**

Page 23: Some part of the data flow should be written somewhere else in a statement at the end of the paper, not in the main part. For example, which platforms are planned to be tested.

**The paragraph was edited as follows:**

**As soon as a larger fraction of iCUPE data products are available, we explore different platforms to disseminate the iCUPE data sets. To facilitate and standardize access to data cloud-based online platforms, known as …**

In summary, I recommend the publication of the manuscript in ACP after major revision. I suggest the authors to read the complete manuscript carefully and try to understand the scientific statements of the co-authors. Please feel free to criticize and ask questions. I understand that this is a huge amount of work, but that would definitely improve the quality of the paper and help non-expert readers to understand the presented results of this very important topic.

**We thank the referee for the encouraging comment! With the joint effort from the authors, we edited the manuscript to improve the quality as a whole.**